# Learning Unified Representations of Normalcy for Time Series Anomaly Detection

## Abstract

The core challenge in unsupervised anomaly detection is identifying abnormal patterns without prior knowledge of their characteristics. While existing methods have addressed aspects of this problem, they often struggle to learn a robust representation of the normal data distribution that is distinct from anomalous patterns. In this paper, we present a novel framework, Unified Unsupervised Anomaly Detection ($U^2$AD), that comprehensively addresses anomaly detection in multivariate time series. Our approach learns the underlying data distribution of normal samples by utilizing score-based generative modeling. We introduce a novel time-dependent score network and a unified training objective that together delineate the manifold of normal data while considering both local and global temporal contexts. Reconstruction is then performed via a deterministic sampling process using an ordinary differential equation solver. Our extensive experimental evaluations demonstrate that $U^2$AD not only outperforms current state-of-the-art methods in detection accuracy but also identifies anomalies at significantly earlier stages of their occurrence.

## 1 Introduction

The foundational challenge in unsupervised anomaly detection is not merely to find outliers, but to construct a definitive model of "normalcy" from data that are assumed to be predominantly normal, yet may contain unlabeled anomalies. An effective model of this kind must learn the complex high-dimensional structure of the normal data-generating process so precisely that any deviation, no matter how subtle, becomes self-evident. However, the prevailing paradigms in time series anomaly detection have been built on simplifying assumptions that limit their robustness.

Existing unsupervised methods have approached this challenge from different, often isolated, perspectives. Reconstruction-based approaches, such as autoencoders (AEs) or variational autoencoders (VAEs), learn a compressed latent representation, assuming the low-dimensional latent variable only contains useful information to reconstruct normal samples (Park et al., 2018; Su et al., 2019). Yet, without contextual information to guide the design of the reconstruction loss, these approaches can be susceptible to errors based on the relative distribution of normal and anomalous samples. Another branch of these methods employs deep generative models to learn the underlying probability distribution of normal data using Generative Adversarial Networks (GANs) (Zhou et al., 2019; Li et al., 2019a). However, their adversarial training process can be notoriously unstable, often leading to mode collapse, a critical weakness that leaves the model vulnerable to failing to detect novel yet normal anomalies. Density-based techniques aim to approximate the probability density distribution of normal data points (Yairi et al., 2017; Zong et al., 2018a), while boundary-based methods (Ruff et al., 2018; Shen et al., 2020) focus on finding a compact hypersphere that encloses the representation of the normal data. Both of these paradigms, however, risk oversimplifying the complex, multi-modal geometry of the underlying data distribution. Even attention-based models (Xu et al., 2021; Dai et al., 2024; Song et al., 2023), while excelling at capturing temporal dependencies, face limitations stemming from their primary focus on reconstruction fidelity. Ultimately, a common weakness across these diverse paradigms is their reliance on a single principle to define normalcy, which leaves them vulnerable to specific types of unseen anomalies.

To overcome these limitations, a solution must be guided by a more comprehensive set of principles. We believe it is essential to explicitly estimate the distribution of non-anomalous data while being

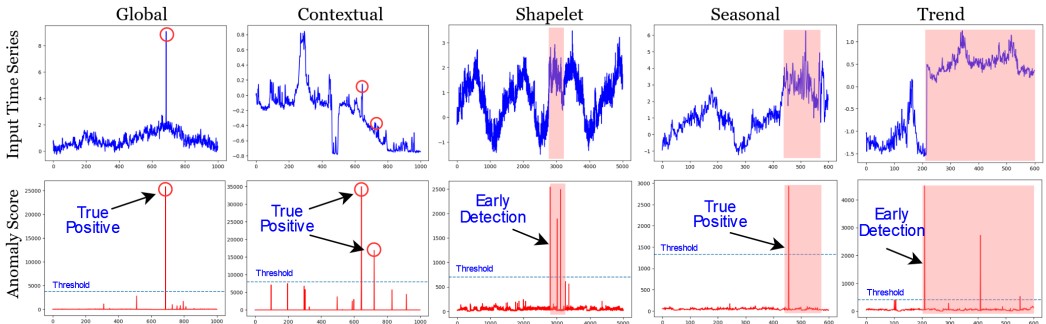

Figure 1: Anomaly detection for different kinds of anomalies, i.e. global, contextual, shapelet, seasonal and trend anomalies (Lai et al., 2021). The top and bottom row figures represent input time series and anomaly scores, respectively. Point anomalies and pattern anomalies are represented by red circles, and red segments respectively. Dotted blue lines in the bottom figures represent threshold.

robust to unlabeled anomalies, and develop a criterion that clearly delineates this normal distribution from anomalous deviations. To realize this hypothesis, we introduce the Unified Unsupervised Anomaly Detection ($U^2AD$) framework. Our work is built on the fundamental premise that a comprehensive model of normalcy can be learned by directly modeling the score function—the gradient of the log-probability density of the data distribution. This approach offers a powerful, non-parametric sampling process to characterize the topography of the data landscape, providing the foundation upon which to build such a criterion.

We also argue that a principled solution must also leverage the characteristics of each data point with respect to its adjacent neighbors as well as the global data pattern. To achieve this, we propose a novel time-dependent, dual-pathway score network explicitly designed to disentangle and model both local point-wise relationships and global series-wide correlations. Finally, we address the need for an accurate reconstruction formulation capable of minimizing errors for non-anomalous data while maximizing them for anomalous data. We accomplish this with a unified training objective that synergistically combines four objectives, each designed to enforce a distinct but complementary aspect of normalcy: score matching to learn the data distribution's gradient field, volume minimization to ensure a compact representation of normal data, contextual information gain to model temporal dependencies, and a reconstruction loss to guarantee fidelity. This is paired with a deterministic ODE solver for stable sample generation. We summarize our contributions as follows:

- We introduce a new paradigm for the detection of unsupervised time series anomalies that unifies the principles of density, boundary, reconstruction, and contextual learning through the powerful lens of score-based modeling.
- We propose a novel dual-pathway score network architecture with a contextual information gain objective, enabling the model to learn rich representations of local and global temporal dependencies.
- Through extensive experiments, we show that ($U^2AD$) not only sets a new state-of-the-art in detection accuracy but also identifies anomalies significantly earlier than existing methods. On average, ($U^2AD$) flagged anomalies before they progressed beyond 38.5% of their total duration, a drastic improvement over prior approaches.

## 2 BACKGROUND

Unsupervised Anomaly Detection (UAD) operates under two key assumptions. First, anomalies are by nature unknown and can be highly diverse, sometimes sharing characteristics with non-anomalous data points. Second, non-anomalous data points significantly outnumber anomalous ones. These assumptions make it infeasible to directly model the distribution of anomalies. Consequently, the primary goal of UAD is to build a robust model of the normal data distribution, $p_{\text{data}}(x)$. Data points that deviate significantly from this modeled distribution are detected as anomalies.

## 2.1 DENOISING SCORE MATCHING

A fundamental assumption in statistical machine learning is that all normal data points form a coherent, albeit potentially complex, underlying probability distribution. Accurately estimating this distribution is therefore essential for anomaly detection. A direct approach to modeling $p_{\text{data}}(x)$ involves parameterizing a density function with any model ($m$). However, most expressive model families define an un-normalized density, $\tilde{p}_m(x;\theta)$; which must be normalized by the partition function, $Z_\theta$, such that $p_m(x;\theta) = \frac{1}{Z_\theta}\tilde{p}_m(x;\theta)$ (Hyvärinen & Dayan, 2005; Song et al., 2020a). While methods using implicit densities like VAEs or GANs bypass this issue, they can struggle to precisely delineate the boundary between normal (in-distribution) and anomalous (near-distribution) data points.

An elegant solution to this problem is to instead model the score function (or score), defined as the gradient of the log-density with respect to the data, $\nabla_x \log p(x)$. The score function's key advantage is its independence from the partition function, since $\nabla_x \log p_m(x;\theta) = \nabla_x \log \tilde{p}_m(x;\theta)$ (Goodfellow et al., 2016; Li et al., 2019b). Intuitively, score represents a vector field that indicates in the direction of the steepest ascent of the data density (Liu et al., 2016). Given any probability function, the score can be easily computed and vice versa.

Denoising score matching (DSM) simplifies the learning objective for the score function by leveraging a perturbation kernel (Vincent, 2011), avoiding the direct computation of high-order derivatives of the data distribution required by other score matching variants. Instead of matching the score of the original data, DSM learns the score of noisy data, which provides better regularization and a more stable training objective (Goodfellow et al., 2016). The objective is to minimize the expected squared L2-norm between the model's score prediction and the true score of the perturbed data distribution:

$$L(\theta) \propto \mathbb{E}_{q_\sigma(\tilde{x}|x)p_d(x)}\left[\|s_m(\tilde{x};\theta) - \nabla_x \log q_\sigma(\tilde{x}|x)\|_2^2\right] \tag{1}$$

Here, $q(\cdot)$ is the perturbation kernel with standard deviation, $\sigma$, and $\tilde{x}$ is the perturbed version of the input $x$.

## 2.2 SCORE-BASED GENERATIVE MODELING

Score-based generative modeling (SGM) generalizes DSM to a continuous spectrum of noise levels ($t$), conceptualized through stochastic differential equations (SDEs). This diffusion process closely resembles the Brownian motion, where the movement of a particle immersed in a fluid appears random due to unpredictable interactions with other particles. Such a process can be mathematically represented as the solution to an Itô SDE using the equation (Gardiner et al., 1985; Øksendal, 2003):

$$d\mathbf{x} = \mathbf{f}(\mathbf{x}, t)dt + g(t)d\mathbf{w} \tag{2}$$

where $\mathbf{f}(\cdot, t) : \mathbb{R}^d \to \mathbb{R}^d$ is the drift coefficient of the SDE, while $g(t) \in \mathbb{R}$ is the diffusion coefficient, and $\mathbf{w}$ signifies the standard Brownian motion. Let's consider, $p_0$ is the data distribution of i.i.d. samples $\mathbf{x}(0) \sim p_0$, and the prior distribution, denoted as $p_T : \mathbf{x}(T) \sim p_T$ possesses a tractable form. The noise introduced by the diffusion process at time step $T$ is sufficiently significant to guarantee that $p_T$ remains independent of $p_0$. Inversely, by initiating from a sample originating in the prior distribution $p_T$ and reversing the diffusion process backwards in time, we can acquire a sample from the data distribution $p_0$. It is defined by the subsequent SDE in reverse time (Anderson, 1982):

$$d\mathbf{x} = [\mathbf{f}(\mathbf{x}, t) - g(t)^2 \nabla_\mathbf{x} \log p_t(\mathbf{x})]dt + g(t)d\bar{\mathbf{w}} \tag{3}$$

Here, $\bar{\mathbf{w}}$ is Brownian motion in the reverse time direction, and the term $\nabla_\mathbf{x} \log p_t(\mathbf{x})$ is the time-dependent score function. To implement this, a time-dependent neural network, $\mathbf{s}_\theta(\mathbf{x}, t)$, is trained to approximate the true score using a continuous version of the DSM objective (Song et al., 2020b):

$$\min_\theta \mathbb{E}_{t \sim \mathcal{U}(0,T)}[\lambda(t)\mathbb{E}_{\mathbf{x}(0) \sim p_0(\mathbf{x})}\mathbb{E}_{\mathbf{x}(t)|\mathbf{x}(0) \sim p_{0t}(\mathbf{x}(t)|\mathbf{x}(0))}$$

$$[\|\mathbf{s}_\theta(\mathbf{x}(t), t) - \nabla_{\mathbf{x}(t)} \log p_{0t}(\mathbf{x}(t) \mid \mathbf{x}(0))\|_2^2]] \tag{4}$$

Here, $\mathcal{U}(0, T)$ represents a uniform distribution spanning the interval $[0, T]$, $p_{0t}(\mathbf{x}(t) \mid \mathbf{x}(0))$ signifies the transition kernel of the forward SDE, and $\lambda(t) \in \mathbb{R}_{>0}$ represents a positive weighting function. While powerful for general data, applying score-based models to time series presents unique challenges due to the complex local and global temporal and contextual dependencies, which our proposed method aims to address.

## 3 METHODOLOGY

### 3.1 PROBLEM FORMULATION

Let $\mathbf{X} \in \mathbb{R}^{L \times d}$ denote an input multivariate time series (MTS) of length $L$ with $d$ features. For anomaly detection, we process windows of this series, $\mathbf{x} \in \mathbb{R}^{N \times d}$, where $N$ is the window size. This input window $\mathbf{x}$ is predominantly composed of normal data points, which we conceptually denote as $\mathbf{x}_n$, but may contain a small subset of anomalies, $\mathbf{x}_a$, due to their unknown nature and patterns. Consistent with the assumptions of UAD (Section 2), the vast majority of observed data points are normal, i.e., $n(\mathbf{x}_n) \gg n(\mathbf{x}_a)$ over the entire dataset. Therefore, the observed data distribution $p_{\text{data}}(\mathbf{x})$ is overwhelmingly dominated by normal samples.

Our proposed $\text{U}^2\text{AD}$ framework learns the score function of this normal data distribution $p_0(\mathbf{x})$. During inference, we utilize the learned score function to guide a reverse diffusion process, effectively reconstructing the input $\mathbf{x}$ as $\hat{\mathbf{x}}$ by sampling from the estimated normal distribution (Figure 2). The deviation of $\mathbf{x}$ from its reconstruction $\hat{\mathbf{x}}$ then contributes to the final anomaly score. A small reconstruction error implies the sample belongs to the normal data manifold, whereas a higher error suggests a significant deviation, indicating an anomaly.

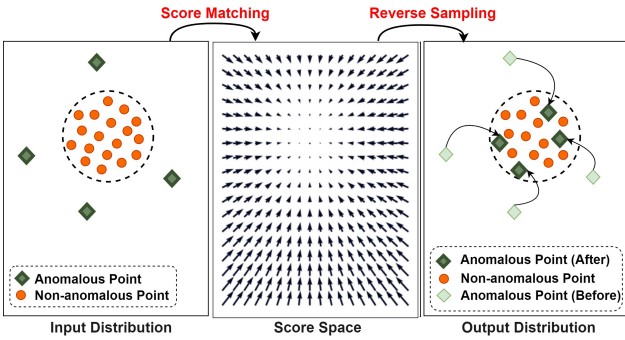

Figure 2: Overview of our proposed framework through denoising score matching and reverse sampling.

To achieve this effective reconstruction and anomaly scoring, we formulate the problem into three interdependent sub-problems. First, we aim to model the score function of the dominating normal data distribution by drawing inspiration from DSM (details in Section 3.2 and 3.3). Second, we address delineating the normal data manifold's boundary by leveraging both local point-wise and global series-wide contextual information, combined with a boundary-enforcing objective, to define a robust and compact representation of normalcy (detailed in Section 3.3 and 3.5). Finally, the third sub-problem focuses on regulated deterministic sample generation from the learned normal distribution for accurate reconstruction. This is achieved by employing SGM and augmenting their sampling process with objectives that enforce the coherence and compactness of the normal data manifold, thereby regulating the generation of in-distribution samples (details in Section 3.4).

### 3.2 DATA PERTURBATION WITH SDE

Motivated by the principles of SGM, we define a forward diffusion process to perturb the data with Gaussian noise over continuous time, following the general form of Eq. 2. This continuous perturbation process enables our framework to learn the score function of the underlying normal data distribution via a weighted DSM objective (Eq. 4). The selection of SDE governs the effectiveness of reverse samplers and sample reconstruction. Song et al. (Song et al., 2020b) proposed three primary SDE variants: variance preserving (VP), variance exploding (VE), and sub-VP. We specifically employ the VP SDE due to its desirable property that the data distribution $p_t(\mathbf{x}(t)|\mathbf{x}(0))$ remains Gaussian at any time step $t$, whose score function $\nabla_{x(t)} \log p_t(\mathbf{x}(t)|\mathbf{x}(0))$ can be precisely estimated. Furthermore, VP-SDE ensures that the magnitude of $\mathbf{x}(t)$ does not explode, leading to more stable training for high-dimensional data. From Eq. 2, if the limit of $dt \to 0$, we get the following equation for the VP SDE:

$$\mathrm{d}\mathbf{x} = -\frac{1}{2}\beta(t)\mathbf{x}\mathrm{d}t + \sqrt{\beta(t)}\mathrm{d}\mathbf{w} \tag{5}$$

Here, $\beta(t)$ is the noise scale function between $\beta_{\text{min}}$ and $\beta_{\text{max}}$. The forward diffusion process transforms an original data window $\mathbf{x}(0) \in \mathbb{R}^{C \times W}$ into a noisy sample $\mathbf{x}(t)$ at time $t \in [0, T]$, where $x(t) \sim p_t(\mathbf{x}(t) \mid \mathbf{x}(0))$.

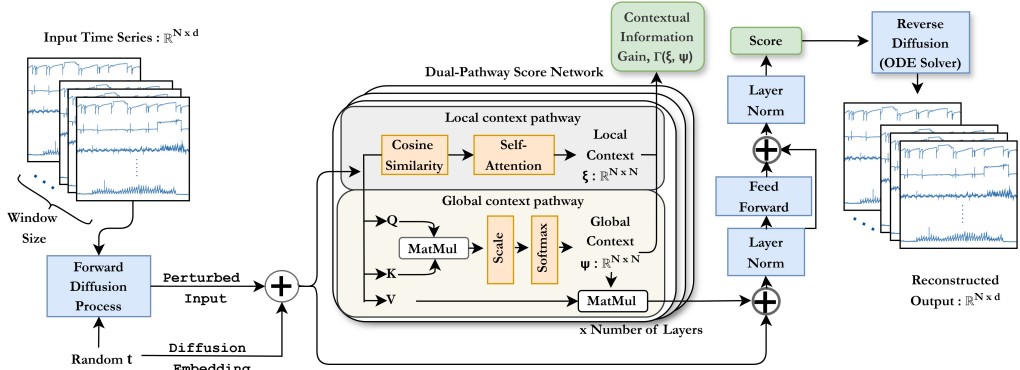

Figure 3: Details of U$^2$AD model.

## 3.3 SCORE MODEL

An ideal score model for anomaly detection must not only learn the score of the dominant, non-anomalous data distribution but also explicitly define its boundaries to distinguish normal variations from true anomalies. To achieve this, we propose a novel score network that incorporates a context-aware boundary regularization term into the training objective, guided by local and global contextual characteristics extracted during the score estimation process.

Given any input of $d$ dimension and $N$ length, the input, $\mathbf{x}(0) : \mathbb{R}^{N \times d}$ is first perturbed with SDE using Eq. 5 to generate a perturbed input, $\mathbf{x}(t) : \mathbb{R}^{N \times d}$, at a random noise level step, $t$. Our dual-pathway score network (Figure 3) takes the perturbed data and the embedded noise scale as input. The network is composed of K stacked transformer-based layers. At each layer k, the model processes the input through two parallel branches to extract contextual characteristics before computing the final score and after the final layer $\mathbf{K}$, the network outputs the final score estimate, $\mathbf{s}_\theta(\mathbf{x}(t), t)$.

**Global Contextual Characteristics** ($\psi : \mathbb{R}^{N \times N}$)**:** The first branch, global context pathway, (Figure 3) utilizes a standard multi-head self-attention mechanism, as seen in transformers (Vaswani et al., 2017). The attention matrix at the k-th layer, $\psi_k : \mathbb{R}^{N \times N}$, captures the long-range, global dependencies as it explicitly encodes the pairwise influence between every point in the sequence. We define the global characteristics as the attention weights themselves: $\psi_k(\mathcal{Q}, \mathcal{K}, \mathcal{V}) = \text{Softmax}(\frac{\mathcal{Q}\mathcal{K}^T}{\sqrt{d_{model}}})$ instead of merely being an intermediate step, we directly leverage, $\psi_k$ as a map of the global data structure, which informs our boundary regularization.

**Local Contextual Characteristics** ($\xi : \mathbb{R}^{N \times N}$)**:** The second branch, local context pathway, is designed to capture fine-grained local patterns. For each time step $i$ in the input sequence $x_k$, we compute the cosine similarity between a single point and every other point within a defined window, producing a similarity matrix. We chose cosine similarity because it is invariant to the magnitude of individual data points, allowing the model to focus on the shape and pattern of local temporal segments, which is often crucial for identifying subtle anomalies. This matrix is processed by an attention layer to learn a weighted aggregation of these local relationships. We define the local characteristic, $\xi^k$ as follows:

$$\mathbf{y}^k = \text{MultiHead}(\Upsilon(\mathbf{x}^k)) \tag{6}$$

$$\xi^k = \text{norm}(\mathbf{x}^k + \text{softmax}(\text{linear}(\mathbf{y}^k))) \tag{7}$$

Here, $\Upsilon$ represents the cosine similarity of data in the same window, and MultiHead denotes a multi-head transformer. Context-aware boundary regularization is achieved by a dual-objective optimization, maximizing boundary based on local characteristics while simultaneously minimizing it based on global characteristics. This methodology promotes the learning of a compact and tightly bounded representation for the dominant (normal) data distribution (Xu et al., 2021). This regularization is specifically formulated to enhance the contextual similarity among samples presumed to belong to the normal class.

### 3.4 DETERMINISTIC REVERSE SOLVER

Having successfully identified the in-distribution and contextual characteristics, the next goal is to generate samples from this score space. We take inspiration from reverse samplers in SGM, specifically the reverse SDE described in Equation 3. However, we wish to reconstruct and regain data close to our original data, which limits the mentioned reverse sampling process as it introduces additional randomness. To overcome this, we leverage the fact that for every diffusion process, there is a corresponding deterministic process, and their trajectories have identical marginal probability densities, as SDE denoted as $p_t(x)$ for $0 \leq t \leq T$.. The deterministic process can be rewritten as an ODE (Maoutsa et al., 2020; Song et al., 2020b).

$$\mathrm{d}x = [\mathbf{f}(x, t) - \frac{1}{2}g(t)^2 \nabla_x \log p_t(x)]\mathrm{d}t \tag{8}$$

We employ this deterministic process, known as probability flow ODE (Song et al., 2020b), to eliminate randomness and stochasticity during the generation of a new sample. Furthermore, this deterministic process often leads to more stable and efficient sampling compared to its stochastic counterpart, without requiring additional trainable parameters for the solver itself. Moreover, we have the flexibility to choose any numerical method for integrating the probability flow ODE backward in time to generate samples. Therefore, if the time series has any tractable form, the sampling process through any solver $\zeta$ is given by $\zeta(\mathbf{s}_\theta(\mathbf{x}(t), t), t)$ which produces the reconstructed output, $\hat{\mathbf{x}}(0) : \mathbb{R}^{N \times d}$. If $\zeta$ is capable of fully reconstructing the non-anomalous part of $\mathbf{x}(0)$, then $||\mathbf{x}_n(0) - \hat{\mathbf{x}}_n(0)|| \to 0$. As the sampling process is driven by the dominating distributions, the anomalous part of the input signal, sampled from the non-anomalous data distribution, will make their reconstruction error higher, $||\mathbf{x}(0) - \hat{\mathbf{x}}(0)|| = ||\mathbf{x}_a(0) - \hat{\mathbf{x}}_a(0)|| > 0$. This serves as a key component for anomaly detection approach.

### 3.5 THE TRAINING OBJECTIVES

Several training objectives have been developed, each of which addresses a particular aspect of our unified AD framework. The formulation and contribution of these objectives are presented below.

**Denoising Score Matching Objective:** Since the noise added to the input belongs to a Gaussian distribution, the denoising score matching (DSM) objectives from Eq. 1 and Eq. 4, can be rewritten by approximating the expectation using empirical means as follows (Vincent, 2011):

$$L_{\mathrm{DSM}}(\theta) = \frac{1}{2N} \sum_{i=1}^{N} \left[ \left\| \mathbf{s}_\theta(\mathbf{x}_{i,:}(t), t) - \frac{\mathbf{x}_{i,:}(t) - \mathbf{x}_{i,:}(0)}{\sigma^2} \right\|_2^2 \right] \tag{9}$$

We reformulate the above equation for each data point of the time series. This objective not only provides insights into high-density regions but also captures the patterns of the data distribution.

**Volume Minimization Objective:** To enforce a strong structural prior on the learned score function, we recast the score vector, $\mathbf{s}_\theta(\mathbf{x}_{i,:}(t), t)$, not merely as a static gradient, but as a representation of the underlying system dynamics at a given point. Inspired by the concept of deep one-class classification (Ruff et al., 2018), we implicitly enforce constraints on the boundary by minimizing volume of the hypersphere that bounds the non-anomalous data in the latent score space. We fix the center of the minimum-energy distribution and make it non-learnable to avoid hypersphere collapse (Ruff et al., 2018). If the center of the data distribution in latent space is $c$, the training objective can be written as, $L_{\mathrm{VM}} = \left[ \|\mathbf{s}_\theta(\mathbf{x}_{i,:}(t), t) - c\|^2 \right]_{i=1,\ldots,N}$. This objective enforces a principle of uniformity, forcing the model to learn the varied patterns of normal data into a single, consistent behavior. By constraining these functions to originate from a compact region in their functional space (represented by proximity to a center $c$), we prevent the model from learning overly complex or disjoint gradient fields that might inadvertently map some anomalies close to the normal manifold.

**Contextual Information Gain:** This objective is built on the hypothesis that for non-anomalous time-series data, local contextual patterns should align with the overarching global structure of the sequence. Anomalies frequently manifest as a localized breakdown in this alignment. To quantify this discrepancy, we employ the symmetric Kullback-Leibler (KL) Divergence, also known as Jeffrey Divergence, for achieving a balance between maximizing the boundary based on local characteristics and minimizing it based on global characteristics (Neal, 2007; Nguyen et al., 2017). Due to the

opposing nature of the two objectives, we also employ a minimax strategy, similar to game theory applications (Grünwald & Dawid, 2004; Xu et al., 2021). Our model is trained for the local context to estimate the global context during the minimization phase, while the global context approximates the local context during the maximization phase. This approach enables the model to effectively adapt to various patterns in the input time series, establishing a trade-off in defining its boundary given by:

$$\Gamma(\xi, \psi) = \frac{1}{K} \sum_{k=1}^{K} \left[ \text{KL} \left( \xi_{i,:}^k \| \psi_{i,:}^k \right) + \text{KL} \left( \psi_{i,:}^k \| \xi_{i,:}^k \right) \right]_{i=1,\ldots,N} \tag{10}$$

where $\text{KL}(\cdot \| \cdot)$ represents KL divergence calculated between two discrete probability distributions associated with each row of $\xi$ and $\psi$ in $k$-th layer. Anomalies exhibit relatively lower resemblance to non-anomalous data, thereby reducing the $\Gamma$ value, while non-anomalous points have higher $\Gamma$. Thus, the objective is to maximize the gain in contextual information to distinguish abnormal patterns.

**Reconstruction Loss:** After successful sampling, we employ the reconstructed output and the original output to derive the reconstruction loss ($L_{\text{Rec}}$), defined as the mean squared error (MSE) function, $\left[ \| \hat{\mathbf{x}}_{i,:} - \mathbf{x}_{i,:} \|^2 \right]_{i=1,\ldots,N}$. While $L_{\text{DSM}}$ guides the model to learn the overall data distribution, we include an explicit reconstruction loss to ensure that the deterministic ODE solver produces an output that is an exact restoration of the non-anomalous components of the original input.

The final training objective combines the above four losses as the following objective:

$$L_{\text{total}} = L_{\text{DSM}}(\theta) + \lambda_1 \times L_{\text{Rec}} + \lambda_2 \times L_{\text{VM}} - \lambda_3 \times \Gamma(\xi, \psi; \mathcal{X}; t) \tag{11}$$

This novel training objective encompasses all aspects of the solution, including information regarding density, and contextual characteristics in addition to implicit boundary specification, and accurate sampling. By combining these four elements, the model learns not just to recognize normal data representation, but to understand its underlying behavior and structure.

### 3.6 MEASURE OF ANOMALY LIKELIHOOD

For anomaly score of any given series, we introduce a novel score-driven distance metric in addition to contextual information gain and reconstruction error. Anomaly score is given by,

$$\left[ \text{Softmax}(-\Gamma(\xi, \psi)) \odot \| \mathbf{x}_{i,:}(0) - \hat{\mathbf{x}}_{i,:}(0) \|^2 + \| \mathbf{s}_\theta(\mathbf{x}_{i,:}(t), t) - c \|^2 \right]_{i=1..N} \tag{12}$$

Here, $\odot$ represents element-wise multiplication, and Anomaly Score($\mathbf{x}$) is the anomaly score for any given data point (more details in Appendix F.2). Anomalies are observed to have low contextual gain, so negating this value assigns them a higher weight. This effectively uses a point's structural inconsistency to scale the importance of its reconstruction error. The final component then identifies any underlying dynamic deviation from the learned norm, creating a robust scoring mechanism.

## 4 EXPERIMENTS

### 4.1 EXPERIMENT SETTINGS

**Datasets:** We assessed the performance of our methodology using four widely recognized, public benchmark datasets for MTS anomaly detection. These real-world datasets, frequently employed in the literature to validate new techniques, include: Mars Science Laboratory (MSL) (Hundman et al., 2018); Soil Moisture Active Passive (SMAP); Secure Water Treatment (SWaT) (Mathur & Tippenhauer, 2016; Goh et al., 2017); and Pooled Server Metrics (PSM) (Abdulaal et al., 2021).

**Baseline Models:** To demonstrate our method's performance, we have chosen 12 state-of-the-art baselines models that cover a diverse range of anomaly detection methods from the literature. These include reconstruction-based models: InterFusion (Li et al., 2021), OmniAnomaly (Su et al., 2019), BeatGAN (Zhou et al., 2019); boundary-based models: DeepSVDD (Ruff et al., 2018); density-based models: DAGMM (Zong et al., 2018a); classic models: Deep Isolation Forest (DIF) (Xu et al., 2023); attention-based models: AnomalyTrans (AT) (Xu et al., 2021), TranAD (Tuli et al., 2022), DCdetector (Yang et al., 2023), SARAD (Dai et al., 2024); diffusion based models: IMdiffusion (Chen et al., 2023); and temporal variation models: TimesNet (Wu et al., 2022). For a fair comparison, we follow the model's respective hyperparameters to produce their results.

Table 1: Performance comparison of our model, U$^2$AD, against baselines using F1-score, ADD, and NRD (as %). The best result for each metric is in **bold**.

| Dataset | MSL | | | SMAP | | | SWaT | | | PSM | | | Average | | |
|---|---|---|---|---|---|---|---|---|---|---|---|---|---|---|---|
| Model | F1 ↑ | ADD ↓ | NRD ↓ | F1 ↑ | ADD ↓ | NRD ↓ | F1 ↑ | ADD ↓ | NRD ↓ | F1 ↑ | ADD ↓ | NRD ↓ | F1 ↑ | ADD ↓ | NRD ↓ |
| DeepSVDD | 80.44 | 189.8 | 79.37 | 69.00 | 408.7 | 71.71 | 84.30 | 536.8 | 74.12 | 92.30 | 148.0 | 66.45 | 81.51 | 320.83 | 72.91 |
| DIF | 43.18 | 202.3 | 89.51 | 60.10 | 612.7 | 80.07 | 83.25 | 650.3 | 97.46 | 86.52 | 253.7 | 93.99 | 68.26 | 429.75 | 90.26 |
| DCDetector | 89.48 | 70.86 | 59.90 | 94.05 | 208.2 | 67.20 | 95.86 | 306.9 | 61.91 | 93.85 | 141.4 | 90.16 | 93.31 | 181.84 | 69.79 |
| TranAD | 88.88 | 109.2 | 63.82 | 69.96 | 373.4 | 56.90 | 85.06 | 811.9 | 98.77 | 91.67 | 110.5 | 64.12 | 83.89 | 428.03 | 70.90 |
| DAGMM | 82.66 | 109.4 | 65.41 | 73.84 | 480.6 | 79.48 | 69.21 | 518.0 | 96.25 | 85.48 | 314.9 | 98.64 | 77.80 | 355.73 | 84.95 |
| OmniAno | 84.27 | 80.62 | 62.16 | 87.86 | 101.7 | 42.87 | 81.12 | 69.84 | 17.54 | 86.28 | 32.86 | 68.83 | 84.88 | 71.26 | 47.85 |
| InterFusion | 90.59 | 52.87 | 48.65 | 89.55 | 375.4 | 86.10 | 86.70 | 156.4 | 41.27 | 84.66 | 29.51 | 75.29 | 87.88 | 153.55 | 62.83 |
| BeatGAN | 79.18 | 81.46 | 60.96 | 87.30 | 293.6 | 76.31 | 70.04 | 621.8 | 96.98 | 91.55 | 174.3 | 79.10 | 82.02 | 292.79 | 78.34 |
| AT | 92.19 | 52.03 | 50.65 | 96.49 | 75.33 | 36.17 | 92.26 | 63.60 | 16.47 | 97.50 | 21.70 | 72.23 | 94.61 | 53.17 | 43.88 |
| ImDiffusion | 84.88 | 49.83 | 47.76 | 88.01 | 79.63 | 37.01 | 78.40 | 72.35 | 20.11 | 92.41 | 28.52 | 76.27 | 85.93 | 57.58 | 45.29 |
| TimesNet | 87.90 | 202.9 | 92.13 | 70.46 | 612.7 | 80.07 | 90.30 | 93.45 | 26.65 | 97.85 | 291.3 | 98.27 | 86.63 | 300.09 | 74.28 |
| U$^2$AD (Ours) | **94.60** | **42.83** | **45.50** | **96.94** | **44.96** | **33.56** | **96.19** | **49.57** | **10.62** | **98.50** | **18.15** | 64.40 | **96.56** | **38.88** | **38.52** |

Table 2: Performance comparison of our model, U$^2$AD, against baselines on AUC-ROC, AUC-PR, VUS-ROC and VUS-PR metrics are reported. The best result for each metric is in **bold**.

| Dataset | MSL | | | | SMAP | | | | SWaT | | | | PSM | | | |
|---|---|---|---|---|---|---|---|---|---|---|---|---|---|---|---|---|
| Model | A$_{ROC}$ | A$_{PR}$ | V$_{ROC}$ | V$_{PR}$ | A$_{ROC}$ | A$_{PR}$ | V$_{ROC}$ | V$_{PR}$ | A$_{ROC}$ | A$_{PR}$ | V$_{ROC}$ | V$_{PR}$ | A$_{ROC}$ | A$_{PR}$ | V$_{ROC}$ | V$_{PR}$ |
| Deep SVDD | 59.33 | 13.63 | 65.48 | 19.25 | 39.14 | 10.19 | 41.38 | 11.40 | 79.77 | 51.38 | 64.01 | 39.20 | 64.10 | 45.35 | 65.56 | 44.94 |
| DIF | 43.06 | 14.82 | 52.31 | 17.17 | 60.42 | 17.24 | 60.98 | 17.59 | 81.95 | 71.07 | 59.13 | 39.93 | 68.84 | 45.33 | 65.30 | 49.35 |
| DCdetector | 54.20 | 11.83 | 67.68 | 16.29 | 44.45 | 10.82 | 43.83 | 15.67 | 60.08 | 47.82 | 55.87 | 42.46 | 76.18 | 53.59 | 74.49 | 51.88 |
| TranAD | 53.10 | 13.65 | 59.28 | 17.88 | 41.29 | 10.34 | 41.99 | 11.62 | 81.79 | **72.56** | 60.41 | 42.26 | 61.33 | 43.32 | 61.40 | 44.94 |
| DAGMM | 47.15 | 15.52 | 49.12 | 14.49 | 54.44 | 16.21 | 51.26 | 16.05 | 71.00 | 55.57 | 66.73 | 52.27 | 62.25 | 36.18 | 61.91 | 37.90 |
| OmniAno | 58.14 | 17.80 | 60.70 | 15.79 | 39.26 | 9.77 | 38.99 | 9.84 | 57.85 | 36.65 | 58.59 | 35.18 | 70.78 | 50.62 | 70.02 | 49.37 |
| InterFusion | 62.25 | 10.69 | 63.98 | 11.20 | 55.34 | 14.20 | 54.29 | 13.71 | 71.13 | 58.47 | 64.62 | 56.30 | 54.55 | 27.85 | 56.16 | 25.48 |
| BeatGAN | 68.09 | 13.11 | 69.02 | 14.13 | 52.31 | 13.92 | 53.20 | 13.98 | 79.73 | 62.72 | 71.49 | 61.95 | 58.22 | 35.27 | 57.51 | 35.77 |
| AT | 59.67 | 12.94 | 65.65 | 17.88 | 41.29 | 10.34 | 41.99 | 11.62 | 81.64 | 72.38 | 58.94 | 38.88 | 77.41 | **55.46** | 72.19 | **52.57** |
| ImDiffusion | 40.21 | 20.79 | 37.27 | 21.00 | 38.06 | 10.15 | 37.16 | 9.93 | 62.22 | 40.18 | 54.17 | 32.24 | 67.74 | 46.61 | 69.98 | 48.20 |
| TimesNet | 57.27 | 14.41 | 65.64 | 21.30 | 45.18 | 11.83 | 47.25 | 13.10 | 23.89 | 8.63 | 26.88 | 9.38 | 59.21 | 38.49 | 66.09 | 48.04 |
| SARAD | 56.02 | 17.09 | 61.11 | 17.48 | 57.32 | 19.06 | 57.56 | 18.67 | 80.75 | 65.29 | 69.14 | 49.77 | 60.37 | 42.44 | 63.70 | 43.87 |
| U$^2$AD (Ours) | **73.14** | **21.86** | **69.21** | **23.98** | **62.24** | **19.12** | **61.44** | **19.92** | **82.63** | 60.56 | **73.22** | **56.61** | **78.22** | 53.66 | **77.02** | 52.55 |

**Evaluation Metrics:** To ensure a comprehensive and robust evaluation of anomaly detection performance, we utilize a combination of standard and advanced metrics. We begin with the conventional point-adjusted metrics, precision, recall (discussed in the Appendix E.1) and F1-score. While widely used, these metrics can be sensitive to precise temporal localization and tend to overstate performance. Therefore, for a more robust evaluation against real-world complexities like continuous, range-wise anomalies and labeling noise, we employ the parameter-free, range-based metric Volume Under the Surface (VUS) (Paparrizos et al., 2022). and threshold-independent metric Area Under the Curve (AUC-ROC and AUC-PR) values to provide a comprehensive and robust measure of model performance. Furthermore, to assess the timeliness of detection, we measure the Average Detection Delay (ADD) (Doshi et al., 2022). Recognizing that ADD does not account for varying anomaly durations, we introduce a novel complementary metric: the Normalized Response Delay (NRD). NRD quantifies the detection delay relative to the total duration of the anomaly. For each anomaly episode $i$, with first detection at $T_i$ and episode start and end times $\tau_{i_s}$ and $\tau_{i_e}$:

$$\text{ADD} = \frac{1}{n}\sum_{i=1}^{n}(T_i - \tau_{i_s}) \quad ; \quad \text{NRD} = \frac{1}{n}\sum_{i=1}^{n}\frac{T_i - \tau_{i_s}}{\tau_{i_e} - \tau_{i_s}}. \tag{13}$$

Lower NRD values indicate faster detection relative to the anomaly's duration, offering a more insightful measure of real-world responsiveness. (A detailed exposition of the metrics in Appendix D)

## 4.2 RESULTS

We evaluate our model, U$^2$AD, against state-of-the-art baselines using both threshold-dependent metrics (Table 1) and threshold-independent, range-based metrics (Table 2). To ensure statistical reliability, the presented results are averaged across 5 runs, each with a different random seed. For brevity, full results for precision, recall, and additional baselines are provided in Appendix E.

Table 3: Ablation Studies for different training objective for our proposed method in four real-world datasets in terms of F1, ADD and NRD (as %). D, R, Γ, V represents $L_{\text{DSM}}$, $L_{\text{Rec}}$, contextual information gain and $L_{\text{VM}}$ objectives respectively.

| Training Obj. | | | | MSL | | | SMAP | | | SWaT | | | PSM | | |
|---|---|---|---|---|---|---|---|---|---|---|---|---|---|---|---|
| D | R | Γ | V | F1 ↑ | ADD ↓ | NRD ↓ | F1 ↑ | ADD ↓ | NRD ↓ | F1 ↑ | ADD ↓ | NRD ↓ | F1 ↑ | ADD ↓ | NRD ↓ |
| ✓ | × | × | × | 68.19 | 136.4 | 87.38 | 84.64 | 418.3 | 73.33 | 79.57 | 494.7 | 88.79 | 81.75 | 173.7 | 91.90 |
| × | ✓ | × | × | 58.87 | 175.4 | 88.85 | 68.38 | 425.7 | 76.87 | 81.50 | 533.4 | 88.99 | 79.51 | 194.2 | 92.94 |
| × | × | ✓ | × | 81.17 | 105.1 | 72.56 | 94.04 | 133.3 | 58.42 | 90.64 | 314.6 | 56.48 | 94.38 | 55.04 | 86.10 |
| × | × | × | ✓ | 52.53 | 161.6 | 85.64 | 59.11 | 513.1 | 77.38 | 82.60 | 529.3 | 86.64 | 83.88 | 170.3 | 90.73 |
| ✓ | ✓ | ✓ | ✓ | **94.60** | **42.83** | **45.50** | **96.94** | **44.96** | **33.56** | **96.19** | **49.57** | **10.62** | **98.50** | **18.15** | **64.40** |

The empirical results demonstrate that U$^2$AD establishes a new state-of-the-art in MTS anomaly detection. As shown in Table 1, our model achieves an average F1-score of 96.56%, a significant improvement over the next-best performer, Anomaly Transformer (94.61%). Beyond aggregate detection performance, U$^2$AD shows superior responsiveness. It records the lowest average Anomaly Detection Delay (ADD) of 38.88 and Normalized Response Delay (NRD) of 38.52%. This NRD score, a key highlight, indicates that our model identifies an anomaly after only 38.52% of the anomalous segment has occurred. This capacity for early detection presents a critical advantage over methods like TimesNet, which, despite a high F1-score on some datasets, has a delayed response with an average NRD of 74.28%. Such rapid identification is crucial for enabling timely mitigation in real-world systems. This robustness is further confirmed by the threshold-independent metrics in Table 2. Our method consistently achieves leading performance on AUC and VUS metrics across most datasets, setting new benchmarks on MSL and SMAP. This strong performance on range-based metrics like VUS, which are less sensitive to labeling noise, validates the effectiveness and reliability of our unified detection framework.

## 4.3 ABLATION STUDIES

In our ablation study, we systematically assess the significance of each component within our training objective. One important observation from a complete analysis of the contribution of each objective function (shown in Table 3) is that when denoising score matching and contextual information gain objectives are integrated with the reconstruction error, the overall performance is notably enhanced (detailed results in Appendix F.1). The addition of the minimum volume objective further enhances the performance, demonstrating state-of-the-art results. This suggests that the minimum volume optimization effectively delineates the boundary of the non-anomalous data distribution, leading to improved anomaly detection. Without utilizing DSM loss, the results suffer, emphasizing the importance of DSM loss in guiding the training process, specially on perturbed data. Furthermore, we examine the contribution of each part of our anomaly likelihood measurement (Eq. 12) in Appendix F.2. In both training objective and anomaly criterion analyses, we observe that when minimum volume and reconstruction are not combined, they produce suboptimal outcomes. Nonetheless, upon combining them with reconstruction error and volume minimization, they attain the highest performance across all four benchmark datasets. Lastly, we explore the sensitivity of our model to different choices of window size, number of layers, choices of $\lambda_3$ in Eq. 11, SDE functions, and score models (Appendix F.3, F.4, H.1, H.2, H.3). These experiments demonstrate the robustness of our approach and provide insights into the optimal choices to maximize performance.

## 5 CONCLUSION

This paper presents U$^2$AD, a unified framework for unsupervised anomaly detection in multivariate time series. Diverging from earlier methods that rely on single-factor analysis, U$^2$AD leverages score-based generative modeling to integrate three crucial perspectives: contextual information, boundary optimization, and data density. We introduce a dual-pathway score network with a unified training objective that significantly improves both anomaly detection accuracy and latency. Our method capitalizes on the gradient of the log data distribution and the distance in the score space to establish a robust anomaly criterion. Extensive experiments demonstrate that U$^2$AD achieves state-of-the-art performance on both threshold dependent and independent metrics for real-world benchmarks, highlighting the power of our unified approach for complex time series analysis.

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

# APPENDIX

## A    RELATED WORK

**Anomaly Detection in Multivariate Time Series:** Given the difficulty in collecting and labeling abnormal observations, supervised learning methods face significant limitations in this context (Fraccaro et al., 2016). Supervised techniques (Rodriguez et al., 2010; Park et al., 2017) require labeled training data and can only identify known types of anomalies, limiting their broader applicability. Consequently, unsupervised approaches are the preferred paradigm for anomaly detection. State-of-the-art unsupervised solutions for multivariate time series anomaly detection can be grouped into several categories:

Reconstruction-based methods operate on the assumption that models trained on normal data cannot accurately reconstruct anomalous samples. Therefore, anomalies are identified by a high reconstruction error. For instance, LSTM-VAE (Park et al., 2018) integrates LSTMs into a variational autoencoder (VAE) to capture temporal dependencies. OmniAnomaly (Su et al., 2019) combines a GRU and a VAE to learn robust time-series representations and employs the Peaks-Over-Threshold (POT) method to set dynamic anomaly thresholds. GAN-based models like MAD-GAN (Li et al., 2019a) and TanoGAN (Bashar & Nayak, 2020) use an architecture with LSTM layers, detecting anomalies via both discriminator and reconstruction losses. USAD (Audibert et al., 2020) employs two autoencoders with an adversarial training scheme to isolate anomalies and accelerate training. MSCRED (Zhang et al., 2019) constructs attention-based ConvLSTM networks for temporal modeling and uses a convolutional autoencoder to reconstruct inter-sensor correlation patterns. Although designed for univariate time series, TadGAN (Geiger et al., 2020) can be extended to the multivariate case, using both reconstruction error and a discriminator for detection.

Density-based methods estimate the probability distribution of normal data, assuming that anomalous points will have a low probability. LOF (Breunig et al., 2000) is a classic method that calculates local density to identify outliers. DAGMM (Zong et al., 2018b) combines an autoencoder for dimensionality reduction with a Gaussian Mixture Model to estimate the density of the data in the latent space. Adaptive-KD (Zhang et al., 2018) uses an adaptive kernel density estimation approach for local density estimation in nonlinear systems.

Boundary-based methods aim to learn a boundary in a representation space that separates normal data from anomalies. OCSVM (Tax & Duin, 2004) uses kernel functions to map data into a high-dimensional feature space and finds a hyperplane that maximizes the margin between normal points and the origin. DeepSVDD (Ruff et al., 2018) uses a neural network to find a minimum-volume hypersphere that encloses most of the normal data in a latent space. Expanding on this, THOC (Shen et al., 2020) extracts multi-scale temporal features using a dilated RNN and builds a hierarchical structure of hyperspheres for each temporal resolution.

Attention-based models have recently shown significant promise in time series analysis. AnomalyTrans (Xu et al., 2021), for example, computes an "association discrepancy" by quantifying the difference between a learned prior-association and the series-association of time points. Through a minimax optimization strategy, this approach enhances the distinguishability between normal and anomalous patterns. Similarly, TranAD (Tuli et al., 2022) uses a two-stage reconstruction process where an initial reconstruction refines attention weights to amplify deviations from the original input, thereby highlighting anomalies. DCdetector (Yang et al., 2023) proposes a dual-branch attention structure that uses contrastive learning to focus on the representational differences between normal and anomalous data. More recently, MEMTO (Song et al., 2023) introduced a memory-guided Transformer with a gated memory module and a two-phase training paradigm to mitigate over-generalization. Subsequently, SARAD (Dai et al., 2024) proposed a spatial association-aware approach that leverages a transformer to learn pairwise inter-feature relationships for anomaly detection and diagnosis.

**Diffusion Methods in Time Series:** Diffusion models have recently gained significant traction for time series analysis, with initial research focusing on forecasting and imputation. This work has explored the use of denoising diffusion probabilistic models (DDPM) (Ho et al., 2020) and score-based generative modeling (Song et al., 2020b). For example, CSDI (Tashiro et al., 2021) employs a probabilistic diffusion model for time series imputation, outperforming deterministic baselines.

Building on DDPMs, TimeGrad (Rasul et al., 2021) applies diffusion models autoregressively to generate future sequences for forecasting, achieving strong performance. To address challenges with limited and noisy data, $D^3$VAE (Li et al., 2022) uses a bidirectional autoencoder (BVAE) with denoising score matching and disentangled latent variables.

There has also been a recent upswing in their application to anomaly detection. ImDiffusion (Chen et al., 2023) is an unconditional generative model that leverages imputation techniques from CSDI to identify anomalous regions. Another approach, MadSGM (Lim et al., 2023), is a score-based generative model featuring a conditional score network and an autoregressive denoising loss. Most recently, MODEM (Zhong et al., 2025) was introduced for nonstationary time series, utilizing a coarse-to-fine diffusion paradigm and a frequency-enhanced network to mitigate false alarms and capture complex temporal patterns.

By integrating attention mechanisms for contextual learning with the generative power of diffusion models, we introduce a novel approach that incorporates boundary conditions to effectively distinguish anomalies. Our method yields a substantial improvement in unsupervised anomaly detection (UAD) accuracy, demonstrating the potential of score-driven, context-guided generative models.

## B  DATASET DETAILS

Presented below are the statistical details of the datasets employed in the experiment.

Table 4

| Benchmarks | Applications | Dimension | Window | #Training | #Test(labeled) | Anomaly Ratio |
|---|---|---|---|---|---|---|
| PSM | Server | 25 | 100 | 132481 | 87841 | 0.278 |
| MSL | Space | 55 | 100 | 58317 | 73729 | 0.105 |
| SMAP | Space | 25 | 100 | 135183 | 427617 | 0.128 |
| SWaT | Water | 51 | 100 | 495000 | 449919 | 0.121 |

## C  IMPLEMENTATION DETAILS:

The proposed $U^2$AD is built on the PyTorch library (Paszke et al., 2019) and trained on an NVIDIA A30 24GB GPU. The model's architecture consists of three layers, each with two transformer branches. It uses a model dimension ($d_{model}$) of 512 and 8 attention heads per transformer to capture intricate temporal dependencies. During training, we use a batch size of 256 and the Adam optimizer (Kingma & Ba, 2014) with an initial learning rate of $10^{-4}$. To ensure robust training, we employ early stopping and an Exponential Learning Rate scheduler with a gamma value of 0.25. The time variable $t$ is sampled uniformly from the range $[\beta_{min}, \beta_{max}]$, where $\beta_{min} = 0.1$ and $\beta_{max} = 20$. All random variables are chosen from a Gaussian distribution. For the total loss function, as defined in Eq. 11, the weights $\lambda_1$ and $\lambda_2$ are set to be inversely proportional to the series length ($N$), i.e., $\lambda_1, \lambda_2 \propto 1/N$. In line with the approach in (Xu et al., 2021), $\lambda_3$ is maintained at a fixed value of 3. The Anomaly Ratio (AR) for each dataset is determined using the Gap Statistic Method (Appendix F.2), with values of 1 for the four datasets. The sampling process is computed using the RK45 ODE solver, implemented in `scipy.integrate.solve_ivp` with a relative and absolute tolerance of $10^{-5}$ (`rtol=1e-5`, `atol=1e-5`). To ensure the statistical reliability of our results, each experiment was run three times, and the mean values are reported for all metrics.

## D  DETAILED EVALUATION METRICS

To ensure a comprehensive and fair comparison of our proposed method with benchmark models, we employ a combination of widely accepted and advanced evaluation metrics, carefully selected to address the unique challenges of time series anomaly detection.

### D.1  POINT-BASED METRICS WITH EVENT ADJUSTMENT

We begin our evaluation using conventional point-based metrics: Precision, Recall, and F1-score. These metrics provide a fundamental assessment of classification accuracy. However, standard

point-wise evaluation can be misleading in time series contexts due to the often-continuous nature of anomalies and the inherent ambiguity of precise anomaly start/end times. To mitigate this, we adopt an event-based adjustment technique, which is standard practice in the field (Xu et al., 2018; Su et al., 2019; Shen et al., 2020; Yang et al., 2023). This method considers an entire anomalous event (range) to be correctly identified as a True Positive if at least one data point within that event is flagged as an anomaly by the model. Conversely, a False Negative occurs if no points within an actual anomaly event are detected. A False Positive is a detected anomaly point that does not fall within any ground truth anomalous event. While this approach can potentially overstate a model's performance by not strictly requiring precise temporal localization (e.g., flagging the very beginning of an anomaly), it remains a widely accepted benchmark for initial performance assessment and comparison across models (Doshi et al., 2022).

### D.2 RANGE-BASED VOLUME UNDER THE SURFACE (VUS) METRIC

Despite their common use, conventional point-based metrics are often ill-suited for evaluating real-world time series anomaly detection. Anomalies frequently manifest as continuous, range-wise events (Wagner et al., 2023), and precise anomaly start and end times can be ambiguous. These metrics also struggle with the misalignment between continuous data and discrete binary labels (Garg et al., 2021). To overcome these challenges, we utilize the Volume Under the Surface (VUS) metric (Paparrizos et al., 2022), a fully parameter-free and range-based approach that offers enhanced robustness to labeling noise and detection delays. VUS operates on a soft-labeling scheme instead of binary ground truth labels ($y_t \in \{0, 1\}$), it transforms them into a continuous series $\tilde{\mathbf{Y}} \in [0, 1]$, where values indicate the degree of anomalousness. This transformation is achieved by applying a buffer around each anomalous range. Specifically, for a set of anomalous ranges $\mathcal{P} = \{p_i = (s_i, e_i)\}$, where $s_i$ and $e_i$ are the start and end timestamps, we define a soft label $\tilde{y}_t$ for each timestamp $t$. This value is calculated using a buffer length $l$, typically set to the median length of the anomalous segments in $\mathcal{P}$. The soft label is highest (1.0) within the actual anomaly range and gradually decreases to 0.0 as the distance from the anomaly increases within the defined buffer zones.

This soft-label series, along with the model's continuous anomaly scores, is then used to redefine the components of the confusion matrix. Instead of binary counts, True Positives (TP), False Positives (FP), True Negatives (TN), and False Negatives (FN) are calculated as cumulative sums of the soft labels and model scores. This allows for a more nuanced and continuous calculation of performance. Using these soft metrics, we compute the threshold-independent Area Under the Receiver Operating Characteristic Curve (AUC-ROC) and Area Under the Precision-Recall Curve (AUC-PR). VUS provides a comprehensive, threshold-agnostic, and robust measure of model performance that better reflects real-world detection capabilities.

### D.3 DETECTION LATENCY METRICS

Timeliness of detection is a critical factor in real-world anomaly detection systems. We assess this using two complementary metrics:

**Average Detection Delay (ADD)** The Average Detection Delay (ADD) (Doshi et al., 2022) quantifies the average time difference between the true start of an anomaly and its first detection by the model. For $n$ anomaly episodes, with $T_i$ as the timestamp of the first detection for the $i$-th episode and $\tau_{i_s}$ as its start time, ADD is defined as:

$$\text{ADD} = \frac{1}{n} \sum_{i=1}^{n} (T_i - \tau_{i_s}).  \tag{14}$$

A lower ADD indicates faster absolute detection.

**Normalized Response Delay (NRD)** Recognizing a limitation of ADD, which treats all delays equally regardless of the anomaly's duration, we introduce a novel complementary metric: the Normalized Response Delay (NRD). NRD quantifies the detection delay relative to the anomaly's total duration, providing a more intuitive measure of responsiveness for anomalies of varying lengths. For $n$ anomaly episodes, with $T_i$ as the timestamp of the first detection, and $\tau_{i_s}$ and $\tau_{i_e}$ as the start

and end times of the $i$-th episode, NRD is defined as:

$$\text{NRD} = \frac{1}{n} \sum_{i=1}^{n} \frac{T_i - \tau_{i_s}}{\tau_{i_e} - \tau_{i_s}}. \tag{15}$$

NRD values range from 0 to 1, where 0 indicates immediate detection at the anomaly's onset, and 1 indicates detection at the very end (or not at all). Lower NRD values consistently indicate faster detection relative to the anomaly's duration, offering a more nuanced and insightful measure of real-world responsiveness. This metric is particularly valuable for comparing performance across datasets with diverse anomaly characteristics.

# E  ADDITIONAL RESULTS

## E.1  RESULTS ON OTHER METRICS

Table 5: Precision (P) and Recall (R) comparison for U$^2$AD and baseline models in four real-world datasets.

| Dataset | MSL | | SMAP | | SWaT | | PSM | |
|---|---|---|---|---|---|---|---|---|
| Model | P ↑ | R ↑ | P ↑ | R ↑ | P ↑ | R ↑ | P ↑ | R ↑ |
| DeepSVDD | 75.76 | 85.75 | 98.39 | 53.13 | 89.80 | 79.43 | 98.21 | 87.06 |
| DIF | 32.05 | 66.16 | 80.27 | 48.04 | 96.68 | 73.11 | 90.03 | 83.28 |
| DCDetector | **93.34** | 85.93 | **98.61** | 89.89 | **98.05** | 93.76 | **99.77** | 88.60 |
| TranAD | 91.42 | 86.47 | 94.88 | 55.41 | 89.17 | 81.32 | 98.63 | 85.65 |
| DAGMM | 91.21 | 75.57 | 84.47 | 65.58 | 80.77 | 60.54 | 90.21 | 81.22 |
| OmniAno | 87.26 | 81.47 | 91.47 | 84.53 | 79.85 | 82.44 | 90.32 | 82.58 |
| InterFusion | 90.02 | 91.17 | 88.65 | 90.47 | 84.54 | 88.97 | 88.36 | 81.26 |
| BeatGAN | 76.33 | 82.25 | 83.67 | 91.25 | 62.35 | 79.89 | 90.54 | 92.58 |
| AT | 91.61 | 92.78 | 93.87 | **99.26** | 85.63 | **100** | 96.85 | 98.15 |
| ImDiffusion | 88.21 | 81.79 | 84.75 | 91.54 | 73.23 | 84.36 | 94.17 | 90.71 |
| TimesNet | 91.56 | 84.54 | 87.29 | 59.07 | 95.44 | 85.69 | 97.48 | 98.22 |
| U$^2$AD (Ours) | 92.14 | **97.21** | 96.63 | 97.24 | 92.84 | 99.78 | 98.06 | **98.93** |

## E.2  MORE BASELINE

In addition to the baselines outlined in Table 1 and Table 2, we conducted experiments with additional baseline networks, including LSTM (Hundman et al., 2018), LSTM-VAE (Park et al., 2018), Tcn-ED (Garg et al., 2021), COUTA (Xu et al., 2022), IsolationForest (Liu et al., 2008) and NCAD (Carmona et al., 2021), to underscore the robustness of our framework. The comparison results with these networks are detailed in Table 6, Table 7 and Table 8.

Table 6: Precision (P), Recall (R) and F1-score (F1) (as %) comparison for U$^2$AD against more baseline models in four real-world datasets. The best result for each metric is in **bold**.

| Dataset | MSL | | | SMAP | | | SWaT | | | PSM | | |
|---|---|---|---|---|---|---|---|---|---|---|---|---|
| Model | P ↑ | R ↑ | F1 ↑ | P ↑ | R ↑ | F1 ↑ | P ↑ | R ↑ | F1 ↑ | P ↑ | R ↑ | F1 ↑ |
| LSTM-VAE | 81.54 | 74.96 | 78.11 | 94.67 | 71.24 | 81.30 | 81.63 | 79.98 | 80.80 | 91.65 | 81.64 | 86.36 |
| Tcn-ED | 91.74 | 88.41 | 89.94 | 94.05 | 55.41 | 69.64 | 88.02 | 85.28 | 86.54 | 86.88 | 89.90 | 88.36 |
| COUTA | 87.84 | 84.90 | 86.01 | 90.18 | 56.24 | 69.28 | **98.93** | 76.06 | 86.01 | **99.70** | 86.41 | 92.58 |
| IsolationForest | 75.18 | 39.12 | 51.46 | 92.87 | 55.94 | 69.82 | 88.25 | 81.02 | 84.48 | 90.04 | 83.28 | 86.52 |
| NCAD | 84.76 | 90.61 | 87.59 | 88.96 | 68.45 | 77.36 | 87.83 | 86.84 | 87.34 | 91.35 | 91.78 | 91.56 |
| U$^2$AD (Ours) | **92.14** | **97.21** | **94.60** | **96.63** | **97.24** | **96.94** | 92.84 | **99.78** | **96.19** | 98.06 | **98.93** | **98.50** |

## E.3  RESULTS WITH LIMITED TRAINING DATA

Furthermore, we evaluate the robustness of our proposed method and several baseline models by training them on only 20% of the available training data. This is done to simulate a scenario with a limited amount of training data available for the model to learn from. Despite this limitation, our method demonstrates superior performance compared to the baseline models, illustrating its ability to

Table 7: NRD (as %) and ADD performance comparison with more baseline models and U²AD in 4 datasets and their respective average. The best result for each metric is in **bold**.

| Dataset | MSL | | SMAP | | SWaT | | PSM | | Average | |
|---|---|---|---|---|---|---|---|---|---|---|
| Model | ADD↓ | NRD↓ | ADD↓ | NRD↓ | ADD↓ | NRD↓ | ADD↓ | NRD↓ | ADD↓ | NRD↓ |
| LSTM-VAE | 168.5 | 75.45 | 107.3 | 46.21 | 72.22 | 23.35 | 235.1 | 94.38 | 124.0 | 63.18 |
| Tcn-ED | 200.8 | 89.53 | 612.7 | 80.07 | 624.1 | 92.51 | 173.5 | 86.15 | 335.7 | 86.18 |
| NCAD | 113.5 | 74.41 | 589.3 | 70.48 | 518.5 | 73.70 | 145.5 | 67.37 | 300.7 | 74.69 |
| COUTA | 174.9 | 78.50 | 379.2 | 57.56 | 635.8 | 94.60 | 168.4 | 82.66 | 423.8 | 78.00 |
| IsolationForest | 163.8 | 79.34 | 376.8 | 59.28 | 504.2 | 73.90 | 315.9 | 94.93 | 309.8 | 80.38 |
| LSTM | 215.7 | 100 | 490.9 | 62.81 | 1560 | 100.7 | 285.8 | 94.67 | 534.5 | 90.62 |
| U²AD (Ours) | **42.83** | **45.50** | **44.96** | **33.56** | **49.57** | **10.62** | **18.15** | **64.40** | **36.59** | **45.42** |

Table 8: Performance comparison of our model, U²AD, against more baselines on threshold-independent AUC-ROC and AUC-PR metrics and fully parameter-free VUS-ROC and VUS-PR metrics are reported. The best result for each metric is in **bold**.

| Dataset | MSL | | | | SMAP | | | | SWaT | | | | PSM | | | |
|---|---|---|---|---|---|---|---|---|---|---|---|---|---|---|---|---|
| Model | $A_{ROC}$ | $A_{PR}$ | $V_{ROC}$ | $V_{PR}$ | $A_{ROC}$ | $A_{PR}$ | $V_{ROC}$ | $V_{PR}$ | $A_{ROC}$ | $A_{PR}$ | $V_{ROC}$ | $V_{PR}$ | $A_{ROC}$ | $A_{PR}$ | $V_{ROC}$ | $V_{PR}$ |
| LSTM | 64.84 | 7.71 | 57.32 | 8.05 | 39.51 | 9.14 | 40.83 | 5.24 | 67.97 | 39.30 | 66.33 | 38.18 | 51.29 | 29.42 | 52.09 | 29.57 |
| LSTM-VAE | 62.54 | 9.87 | 61.55 | 9.47 | 43.13 | 10.61 | 44.17 | 9.24 | 69.66 | 44.00 | 70.41 | 48.35 | 64.26 | 26.23 | 62.14 | 32.04 |
| TcnED | 53.50 | 13.97 | 59.60 | 18.15 | 39.42 | 10.26 | 41.93 | 11.41 | 82.42 | **74.43** | 66.46 | 49.36 | 63.68 | 39.39 | 64.42 | 43.36 |
| Couta | 52.21 | 14.91 | 59.32 | 17.88 | 37.60 | 9.80 | 40.72 | 10.85 | **84.14** | 73.11 | 72.75 | 55.04 | 71.21 | **54.89** | 74.11 | **58.14** |
| Isolation Forest | 54.41 | 12.27 | 63.71 | 17.99 | 59.43 | 16.36 | 58.71 | 16.49 | 74.33 | 51.76 | 71.13 | 52.24 | 74.81 | 52.72 | 74.66 | 52.87 |
| NCAD | 56.78 | 10.46 | 49.26 | 11.34 | 40.03 | 9.82 | 41.58 | 10.51 | 58.34 | 45.43 | 56.94 | 46.00 | 67.34 | 48.91 | 68.70 | 47.42 |
| U²AD (Ours) | **73.14** | **21.86** | **69.21** | **23.98** | **62.24** | **19.12** | **61.44** | **19.92** | 82.63 | 60.56 | **73.22** | **56.61** | **78.22** | 53.66 | **77.02** | 52.55 |

accurately learn the density of the non-anomalous data distribution and generalize well to new data. Notably, we do not remove any data from the test set during this evaluation, further showcasing the effectiveness of our method in identifying anomalies in a real-world scenario. Table 9 presents the performance comparison of the baseline models with U²AD on limited training data

Table 9: Quantitative results with limited training data (20%) for four real-world datasets. The P, R, F1, ADD and NRD represent the precision (as %), recall (as %), F1-score (as %), average detection delay, and normalized response delay (as %) respectively. P, R, and F1 are computed using point adjustment strategy. The top-performing methods in each category have been highlighted in bold.

| Dataset | MSL | | | | | SMAP | | | | |
|---|---|---|---|---|---|---|---|---|---|---|
| Model | P | R | F1 | ADD | NRD | P | R | F1 | ADD | NRD |
| TimesNet | 90.99 | 87.82 | 89.38 | 93.14 | 58.13 | 86.78 | 59.07 | 70.29 | 368.4 | 52.67 |
| Tcn-ED | 86.83 | 87.88 | 87.35 | 123.7 | 76.45 | 95.96 | 55.26 | 70.13 | 373.8 | 58.06 |
| NCAD | 88.96 | 75.13 | 81.47 | 114.1 | 64.59 | 76.09 | 70.46 | 73.16 | 676.1 | 76.12 |
| COUTA | 72.87 | 90.87 | 80.88 | 120.9 | 72.68 | **99.63** | 52.98 | 69.17 | 378.3 | 62.00 |
| DCdetector | 92.43 | 76.38 | 83.64 | 105.1 | 73.42 | 99.08 | 83.25 | 90.48 | 265.9 | 76.45 |
| IsolationForest | 91.00 | 79.51 | 84.87 | 67.44 | 48.26 | 90.25 | 56.16 | 69.24 | 378.0 | 58.80 |
| DeepIsolationForest | 84.94 | 41.31 | 55.58 | 199.4 | 89.42 | 76.53 | 54.14 | 63.42 | 547.3 | 77.42 |
| LSTM | **94.02** | 75.20 | 83.57 | 121.5 | 67.83 | 90.27 | 49.87 | 64.25 | 540.8 | 67.42 |
| TranAD | 83.28 | 87.77 | 85.47 | 138.8 | 79.60 | 94.38 | 55.41 | 69.83 | 378.6 | 58.34 |
| DeepSVDD | 87.30 | 87.10 | 87.20 | 111.7 | 68.87 | 99.14 | 53.14 | 69.19 | 408.8 | 71.90 |
| Anomaly Transformer | 91.57 | 96.76 | 94.09 | 44.56 | 39.97 | 93.57 | **99.17** | 96.29 | **68.67** | **30.84** |
| U²AD (Ours) | 91.54 | **98.20** | **94.75** | 38.83 | **38.15** | 96.94 | 96.88 | **96.91** | 106.1 | 47.82 |
| Dataset | SWaT | | | | | PSM | | | | |
| Model | P | R | F1 | ADD | NRD | P | R | F1 | ADD | NRD |
| TimesNet | 95.49 | 84.95 | 89.91 | 414.1 | 57.48 | 97.26 | 98.17 | 97.71 | 59.51 | **44.51** |
| Tcn-ED | 89.34 | 81.94 | 85.48 | 841.4 | 98.00 | 96.74 | 88.80 | 92.60 | 121.1 | 67.84 |
| NCAD | 94.94 | 81.52 | 87.72 | 511.6 | 74.48 | 90.26 | 70.03 | 78.86 | 169.6 | 72.12 |
| COUTA | **99.16** | 73.57 | 84.47 | 983.6 | 96.58 | 66.30 | 96.15 | 78.48 | 211.1 | 92.77 |
| DCdetector | 97.53 | 93.39 | 95.41 | 228.3 | 51.62 | 80.60 | **99.51** | 89.07 | 171.4 | 95.21 |
| IsolationForest | 100 | 65.72 | 79.32 | 647.5 | 97.46 | 88.25 | 81.02 | 84.48 | 51.22 | 73.67 |
| DeepIsolationForest | 98.81 | 69.68 | 81.72 | 762.8 | 94.96 | 96.84 | 82.24 | 88.95 | 257.8 | 98.18 |
| LSTM | 76.51 | 82.34 | 79.32 | 840 | 98.83 | 97.02 | 75.61 | 84.98 | 66.29 | 81.33 |
| TranAD | 93.46 | 73.75 | 82.44 | 780.4 | 97.83 | 98.04 | 85.92 | 91.58 | 110.5 | 64.12 |
| DeepSVDD | 26.58 | 76.88 | 39.50 | 1560 | 100.00 | **99.26** | 80.24 | 88.74 | 199.9 | 88.60 |
| Anomaly Transformer | 90.24 | **100** | 94.87 | 59.69 | 15.24 | 97.02 | 98.30 | 97.66 | 17.68 | 69.08 |
| U²AD (Ours) | 92.23 | **100** | **95.96** | 59.34 | **15.09** | 97.43 | 98.42 | **97.92** | 17.55 | 64.57 |

### E.4 Comparison of Raw Score Model

In the following experiment, we aimed to assess the robustness of our raw score model. The raw score model is configured in a way that the input corresponds to the unperturbed data, and the output represents the reconstructed output. This implies that we do not utilize the score-based generative modeling and its respective training objectives as well as the anomaly criterion. Remarkably, even under these conditions, our raw score model surpasses the performance of the most effective baseline models. A detailed comparison is provided in Table 10.

Table 10: Quantitative results for our raw score model (repurposed to take unpertubed data as input and output the reconstructed input) in four real-world datasets. The P, R and F1 represent the precision, recall and F1-score (as %) respectively. The top-performing methods in each category have been highlighted in bold.

| Dataset | MSL | | | SMAP | | | SWaT | | | PSM | | |
|---|---|---|---|---|---|---|---|---|---|---|---|---|
| Model | P | R | F1 | P | R | F1 | P | R | F1 | P | R | F1 |
| ImDiffusion | 88.21 | 81.79 | 84.88 | 84.75 | 91.54 | 88.01 | 73.23 | 84.36 | 78.40 | 94.17 | 90.71 | 92.41 |
| AnomalyTrans | 91.61 | 92.78 | 92.19 | 93.87 | **99.26** | 96.49 | 85.63 | **100** | 92.26 | 96.85 | 98.15 | 97.50 |
| Ours (Raw Model) | **92.16** | **97.35** | **94.68** | **96.85** | 97.60 | **97.22** | **97.35** | 92.28 | **94.75** | **97.39** | **98.93** | **98.15** |

## F MORE ABLATION STUDIES

### F.1 TRAINING OBJECTIVE

In Table 11, we present the ablation studies of the four training objectives. We experimented with all 15 combinations of the training objectives. Note that, during each experiment, we only employ their respective term(s) in the anomaly criterion (Eq. 12), in addition to reconstruction error. From the results in Table 11, it can be noticed that reconstruction and volume minimization objective work towards the same goal. When they are utilized individually with reconstruction error and DSM loss, they achieve high metric results. However, the best result is found when they are utilized together. The study highlights the significant role of the DSM loss. Without utilizing DSM loss, the results suffer, indicating that other training objectives do not perform well on perturbed data. This emphasizes the importance of DSM loss in guiding the training process, specially on perturbed data. Denoising score matching is the training objective that binds all the other objectives together. This suggests that DSM loss plays a unifying role in the training process.

Table 11: Ablation studies for different training objective. Here, D, R, $\Gamma$, V represents denoising score matching ($L_{\text{DSM}}$), reconstruction ($L_{\text{Rec}}$), contextual information gain ($\Gamma$) and volume minimization ($L_{\text{VM}}$) objectives respectively. For the following experiments, we exclusively employ their respective term(s) in anomaly criterion (Eq. 12), in addition to reconstruction error.

| Training Obj. | | | | MSL | | | SMAP | | | SWaT | | | PSM | | |
|---|---|---|---|---|---|---|---|---|---|---|---|---|---|---|---|
| D | R | $\Gamma$ | V | F1 ↑ | ADD ↓ | NRD ↓ | F1 ↑ | ADD ↓ | NRD ↓ | F1 ↑ | ADD ↓ | NRD ↓ | F1 ↑ | ADD ↓ | NRD ↓ |
| ✓ | × | × | × | 68.19 | 136.4 | 87.38 | 84.64 | 418.3 | 73.33 | 79.57 | 494.7 | 88.79 | 81.75 | 173.7 | 91.90 |
| × | ✓ | × | × | 58.87 | 175.4 | 88.85 | 68.38 | 425.7 | 76.87 | 81.50 | 533.4 | 88.99 | 79.51 | 194.2 | 92.94 |
| × | × | ✓ | × | 81.17 | 105.1 | 72.56 | 94.04 | 133.3 | 58.42 | 90.64 | 314.6 | 56.48 | 94.38 | 55.04 | 86.10 |
| × | × | × | ✓ | 52.53 | 161.6 | 85.64 | 59.11 | 513.1 | 77.38 | 82.60 | 529.3 | 86.64 | 83.88 | 170.3 | 90.73 |
| ✓ | ✓ | × | × | 42.14 | 179.1 | 87.92 | 76.00 | 436.6 | 77.16 | 82.41 | 529.2 | 85.23 | 80.83 | 166.5 | 92.15 |
| ✓ | × | ✓ | × | 52.59 | 167.7 | 82.01 | 83.45 | 341.7 | 70.93 | 93.29 | 272.3 | 53.92 | 93.64 | 94.86 | 86.03 |
| ✓ | × | × | ✓ | 86.54 | 77.61 | 56.42 | 78.08 | 343.1 | 63.55 | 83.82 | 456.3 | 76.70 | 93.64 | 127.0 | 72.64 |
| × | ✓ | ✓ | × | 60.43 | 142.8 | 80.44 | 82.68 | 382.3 | 80.68 | 89.99 | 297.5 | 59.60 | 93.06 | 75.75 | 86.57 |
| × | ✓ | × | ✓ | 50.70 | 173.0 | 85.07 | 67.97 | 512.6 | 80.90 | 80.44 | 482.5 | 89.75 | 80.61 | 218.6 | 95.30 |
| × | × | ✓ | ✓ | 68.89 | 100.8 | 68.20 | 94.36 | 96.87 | 51.59 | 91.69 | 280.9 | 51.35 | 91.86 | 56.10 | 85.57 |
| ✓ | ✓ | ✓ | × | **94.83** | **41.25** | **45.33** | 95.99 | 65.12 | 44.69 | 94.75 | 84.51 | 21.55 | 98.21 | 28.56 | 69.05 |
| ✓ | ✓ | × | ✓ | 85.05 | 83.42 | 61.23 | 70.58 | 400.4 | 68.89 | 84.92 | 478.2 | 74.61 | 91.34 | 142.1 | 73.55 |
| ✓ | × | ✓ | ✓ | 93.46 | 49.69 | 46.20 | 95.77 | 64.67 | 40.67 | 96.02 | 78.51 | 19.77 | 97.82 | 26.90 | 68.64 |
| × | ✓ | ✓ | ✓ | 63.41 | 129.3 | 81.55 | 91.65 | 229.3 | 67.72 | 89.82 | 284.4 | 51.51 | 91.82 | 75.89 | 87.94 |
| ✓ | ✓ | ✓ | ✓ | 94.60 | 42.83 | 45.50 | **96.94** | **44.96** | **33.56** | **96.19** | **49.57** | **10.62** | **98.50** | **18.15** | **64.40** |

### F.2 ANOMALY CRITERION

We conducted experiments with various criteria to determine the optimal anomaly score likelihood. These experiments incorporated a spectrum of anomaly score functions, such as Contextual Information Gain ($\Gamma$), Reconstruction Error, denoising score matching ($L_{\text{DSM}}$), and distance from the

hypersphere center. We listed four different combinations, and the experimental results are presented in Table 12.

1. $\|x_i - \tilde{x}_i\|^2$
2. $\|s_\theta(\tilde{x}_i) - c\|^2$
3. $\text{Softmax}(-\Gamma(\xi, \psi)) \odot \|x_i - \tilde{x}_i\|^2$
4. $\text{Softmax}(-\Gamma(\xi, \psi)) \odot \|x_i - \tilde{x}_i\|^2 + \|s_\theta(\tilde{x}_i) - c\|^2$

Throughout our experiments, we observed distinct performance trends among various Anomaly Criteria. Anomaly Criterion-1 is standard reconstruction error, and Anomaly Criterion-2 is volume minimization error (Ruff et al., 2018). Anomaly Criterion-3 is similar to criterion from Anomaly Transformer (Xu et al., 2021). Notably, during the initial iterations, Anomaly Criterion-3 demonstrated superior performance compared to others, while in the final epochs, when the model's performance plateaued, Anomaly Criterion-4 (Ours) outperformed its counterparts. Our analysis of different anomaly criteria reveals that other anomaly criteria performance lacks consistency across all experiments. In contrast, incorporating multiplication of contextual information gain with the distance from the center consistently produces higher metric results in all experiments. Additionally, we find that relying solely on the reconstruction error or SVDD is insufficient for distinguishing anomalies. The contextual information gain offers valuable insights into temporal correlations and statuses, and a similar trend is observed for the distance from the center. Incorporating characteristic information mitigates the influence of extreme values and balances the terms within the model.

Table 12: Ablation Studies for different training objective for our proposed method in four real-world datasets. F1, ADD and NRD represent the F1-score (as %), average detection delay and normalized response delay (as %) respectively. The top-performing methods in each category have been highlighted in bold.

| Dataset | MSL | | | SMAP | | | SWaT | | | PSM | | |
|---------|-----|-----|-----|------|-----|-----|------|-----|-----|-----|-----|-----|
| Criterion No. | F1 | ADD | NRD | F1 | ADD | NRD | F1 | ADD | NRD | F1 | ADD | NRD |
| 1 | 76.49 | 109.2 | 73.26 | 74.13 | 449.0 | 78.31 | 82.37 | 521.6 | 88.85 | 86.00 | 165.7 | 88.73 |
| 2 | 78.29 | 114.7 | 75.45 | 74.03 | 452.6 | 77.84 | 85.49 | 487.0 | 86.99 | 82.88 | 175.2 | 90.92 |
| 3 | 94.58 | **42.83** | **45.50** | 96.35 | **44.96** | **33.56** | 95.50 | 49.57 | **10.62** | **98.50** | **18.15** | 64.80 |
| 4 | **94.60** | 42.83 | 45.50 | **96.94** | 44.96 | 33.56 | **96.19** | 49.57 | 10.62 | **98.50** | **18.15** | **64.40** |

## F.3 CHOICE OF SDE

Motivated by the success of Variance Preserving (VP) SDE (Eq. 5), Song et al. (Song et al., 2020b) introduced a variant of SDE known as sub-VP SDE. The underlying equation of sub-VP SDE is given by

$$\mathrm{dx} = -\frac{1}{2}\beta(t)\mathbf{x}\mathrm{d}t + \sqrt{\beta(t)\left(1 - e^{-2\int_0^t \beta(s)ds}\right)}\mathrm{d}\mathbf{w} \tag{16}$$

In addition to the standard VP-SDE formulation, it is also possible to derive a version of the SDE, called Variance Exploding (VE) SDE, in the exploding variance setting, when $t \to \infty$. VE-SDE is given by the following equation,

$$\mathrm{dx} = \sqrt{\frac{\mathrm{d}[\sigma^2(t)]}{\mathrm{d}t}}\mathrm{d}\mathbf{w} \tag{17}$$

The results in Table 13 suggest that while VE SDE achieves slightly poor results compared to VP SDE in terms of F1 score, VP SDE detects anomalies at earlier stages. The sub-VP SDE exhibits strong performance in relatively simpler and smaller datasets (e.g., PSM, SMAP). However, its effectiveness diminishes when applied to larger datasets with complex anomalies, such as SWaT, and MSL.

## F.4 SCORE MODEL

In recent literature, U-Net (Ronneberger et al., 2015) and its variants have been employed as the reverse model in diffusion tasks (Li et al., 2022). However, our objective necessitates identifying correlations among both local and global data points, making transformer models the most suitable

Table 13: Quantitative results for $U^2AD$ (Ours) in four real-world datasets. The P, R and F1 represent the precision, recall and F1-score (as %) respectively. The top-performing methods in each category have been highlighted in bold.

| Dataset | MSL | | | SMAP | | | SWaT | | | PSM | | |
|---------|------|------|------|------|------|------|------|------|------|------|------|------|
| SDE Function | F1 | ADD | NRD | F1 | ADD | NRD | F1 | ADD | NRD | F1 | ADD | NRD |
| sub-VP SDE | 76.04 | **41.19** | 45.91 | 95.83 | 74.46 | 42.11 | 81.69 | 559.4 | 94.17 | 95.21 | 43.63 | 71.91 |
| VE SDE | 93.10 | 53.12 | 50.80 | 96.12 | 65.88 | 44.08 | **96.50** | 165.1 | 36.68 | 98.13 | 18.44 | 71.80 |
| VP SDE (Ours) | **94.60** | 42.83 | **45.50** | **96.94** | 44.96 | 33.56 | 96.19 | **49.57** | **10.62** | **98.50** | **18.15** | **64.40** |

candidates for the score model. Moreover, we aim to generate supplementary information during training, much like AnomalyTrans (Xu et al., 2021), where prior and series associations were used to produce information gain, leading to substantial improvements. In our case, the score model requires the embedded noise level and the perturbed signal as inputs, and outputs a vector field of scores. Considering our requirements and motivated by the success of self-attention and transformer models (Vaswani et al., 2017), we opted for a two-transformer network approach for each layer, which produces similar additional information i.e. local and global contextual characteristics and supports our requirements. Our score model not only offers a feasible reconstruction function but also furnishes extra contextual information that can be integrated into the boundary constraints. This way, we can better fulfill our objectives and enhance the precision of anomaly detection.

To this end, we experiment the robustness of our score model and similarity block. We repurpose AnomalyTrans so that it takes perturbed data as input and produces score of the data distribution. The results in Table 14 demonstrate that our proposed time-dependent score model provides superior results.

Table 14: Quantitative results in four real-world datasets for $U^2AD$ (Ours) and repurposed Anomaly-Trans model for score generation . The P, R and F1 represent the precision, recall and F1-score (as %) respectively. The top-performing methods in each category have been highlighted in bold.

| Dataset | MSL | | | SMAP | | | SWaT | | | PSM | | |
|---------|------|------|------|------|------|------|------|------|------|------|------|------|
| Model | P | R | F1 | P | R | F1 | P | R | F1 | P | R | F1 |
| AnomalyTrans | **92.67** | 95.03 | 93.83 | 93.59 | **99.01** | 96.22 | **95.16** | 96.31 | 95.73 | 97.42 | 98.79 | 98.10 |
| $U^2AD$ (Ours) | 92.14 | **97.21** | **94.60** | **96.63** | 97.24 | **96.94** | 92.84 | **99.78** | **96.19** | **98.06** | **98.93** | **98.50** |

## G  VISUALIZATION

In addition to detecting various types of anomalies (Figure 1), we display $U^2AD$'s performance on AD for each of the four benchmark datasets in Figure 4. We exhibit excellent performance not only proper detection, but also early detection of anomalies. We also demonstrate our method's training objective convergence in Figure 5.

## H  PARAMETER SENSITIVITY

### H.1  CHOICE OF $\lambda_3$

$\lambda_3$ serves as the multiplier for contextual information gain, denoted as $\Gamma$, in the final training objective (Eq. 11). AnomalyTrans (Xu et al., 2021) determined the optimal value for this multiplier through their association learning to be 3. In our pursuit of identifying the ideal value for $\lambda_3$ within our training objective, we experimented with values of $1, 2, 3, 4$. While $\lambda_3$ values of 2 and 3 exhibit comparable performance, a comprehensive consideration of F1-score, ADD, and NRD values in our experiments led us to choose $\lambda_3$ as 3. The detailed results are provided in Table 15.

### H.2  CHOICE OF WINDOW SIZE $N$

We conduct experiments with various window sizes for the input data. The selection of the window size is important for effective local and global characteristic learning. After careful evaluation, the

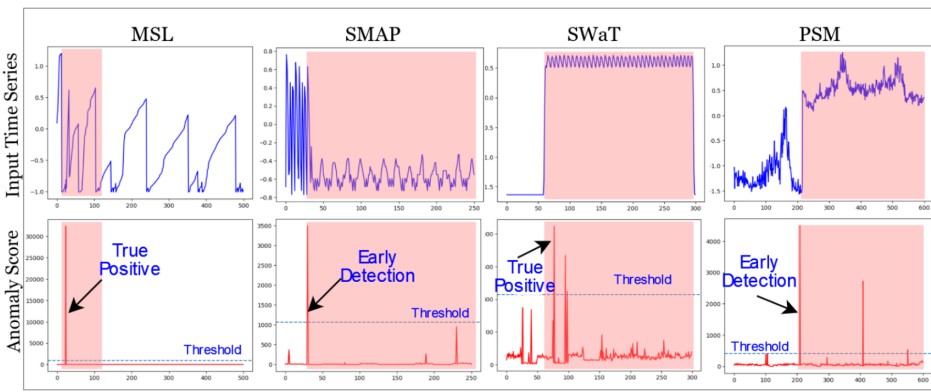

Figure 4: Anomaly detection results for 4 benchmark datasets. The figures in the top row depict input time series, while those in the bottom row show anomaly scores. The dotted blue lines in the bottom row figures denote the thresholds for their respective cases.

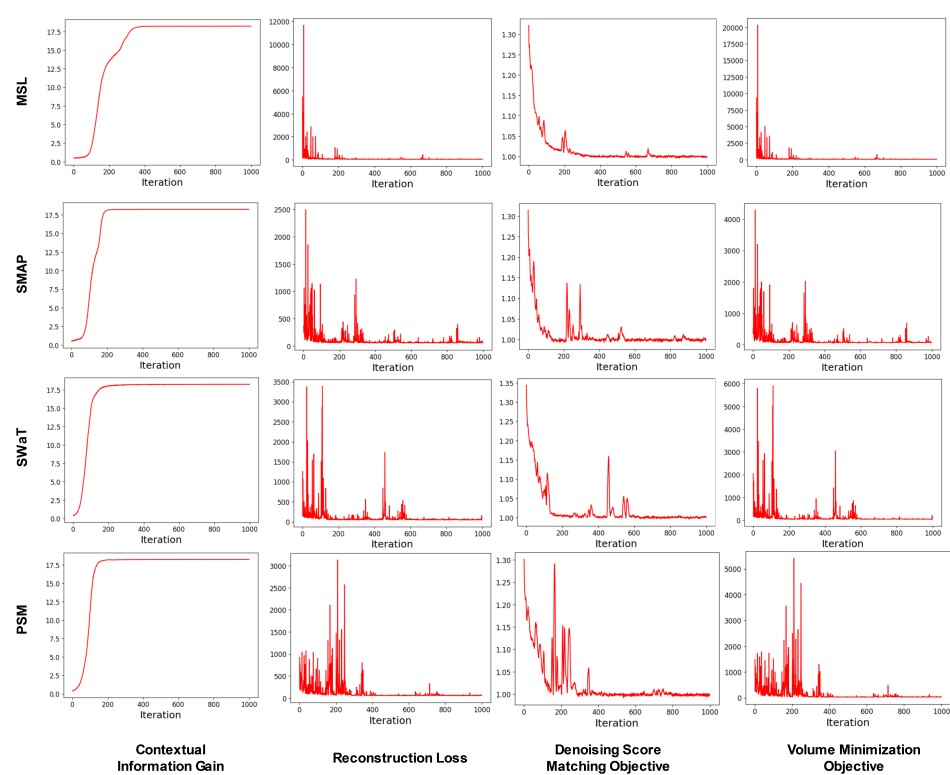

Figure 5: Convergence of different training objective for four benchmark datasets

optimal performance is observed with a window size of 100. This suggests that the learning of characteristics is most effective when employing a medium sized window. Larger window size affects proper learning because of the high length of the input data, while smaller window size does not provide enough context to learn effectively. The detailed results can be found in Table 16.

### H.3 CHOICE OF LAYER NUMBER $K$

The number of layers, $K$ in the score network provides information about the depth of the network. We investigated the influence of varying layer counts, i.e. $K \in \{2, 3, 4\}$. While $K = 4$ performs marginally superior than $K = 3$, it results in increased training and inference times. With this results,

Table 15: Quantitative results for the choice of multiplier, $\lambda_3$, in the training objective (Eq. 11) in four real-world datasets for our method. The P, R, F1, ADD and NRD represent the precision (as %), recall (as %), F1-score (as %), average detection delay, and normalized response delay (as %) respectively. P, R, and F1 are computed using point adjustment strategy. The top-performing methods in each category have been highlighted in bold.

| Dataset | MSL | | | | | SMAP | | | | |
|---|---|---|---|---|---|---|---|---|---|---|
| $\lambda_3$ | P | R | F1 | ADD | NRD | P | R | F1 | ADD | NRD |
| 1 | 92.11 | 96.24 | 94.13 | 46.72 | 44.32 | 93.75 | 99.25 | 96.42 | **44.37** | 34.83 |
| 2 | 91.84 | **97.35** | 94.51 | **41.33** | **38.17** | 93.70 | **99.34** | 96.44 | 48.55 | 34.45 |
| 3 | 92.14 | 97.21 | **94.60** | 42.83 | 45.50 | **96.63** | 97.24 | **96.94** | 44.96 | 33.56 |
| 4 | **93.32** | 86.11 | 89.57 | 60.33 | 51.59 | 93.83 | 98.14 | 95.94 | 55.45 | **33.39** |
| Dataset | SWaT | | | | | PSM | | | | |
| $\lambda_3$ | P | R | F1 | ADD | NRD | P | R | F1 | ADD | NRD |
| 1 | 92.81 | **100** | 96.27 | 56.57 | 15.22 | 97.42 | 98.48 | 97.95 | **16.26** | 71.23 |
| 2 | 93.17 | **100** | **96.46** | 48.77 | 13.24 | 97.43 | **98.95** | 98.19 | 18.75 | 66.44 |
| 3 | 92.84 | 99.78 | 96.19 | 49.57 | **10.62** | **98.06** | 98.93 | **98.50** | 18.15 | **64.40** |
| 4 | 93.02 | 99.82 | 96.30 | 53.66 | 13.18 | 97.26 | 98.92 | 98.08 | 20.38 | 71.84 |

Table 16: Quantitative results for the choice of window size $N$ in four real-world datasets for our method. The P, R, F1, ADD and NRD represent the precision (as %), recall (as %), F1-score (as %), average detection delay, and normalized response delay (as %) respectively. P, R, and F1 are computed using point adjustment strategy. The top-performing methods in each category have been highlighted in bold.

| Dataset | MSL | | | | | SMAP | | | | |
|---|---|---|---|---|---|---|---|---|---|---|
| Window Size $N$ | P | R | F1 | ADD | NRD | P | R | F1 | ADD | NRD |
| 64 | 91.79 | 93.52 | 92.65 | 48.39 | 45.81 | 93.63 | **98.02** | 95.78 | 50.60 | 38.97 |
| 100 | 92.14 | 97.21 | 94.60 | **42.83** | 45.50 | **96.63** | 97.24 | **96.94** | **44.96** | **33.56** |
| 200 | **92.15** | **97.60** | **94.80** | 46.19 | **39.78** | 94.19 | 97.56 | 95.84 | 65.66 | 43.81 |
| Dataset | SWaT | | | | | PSM | | | | |
| Window Size $N$ | P | R | F1 | ADD | NRD | P | R | F1 | ADD | NRD |
| 64 | 91.84 | **100** | 95.75 | 53.17 | 11.41 | 96.93 | 98.41 | 97.67 | 23.85 | 71.78 |
| 100 | 92.84 | 99.78 | 96.19 | **49.57** | **10.62** | **98.06** | **98.93** | **98.50** | 18.15 | 64.40 |
| 200 | **95.06** | **100** | **97.47** | 90.69 | 23.37 | 97.63 | 98.00 | 97.82 | 24.42 | 77.02 |

we trade-off enhanced performance for less training and inference time, and select $K = 3$. The results are shown in Table 17.

Table 17: Quantitative results for the choice of layer numbers, $K$ in four real-world datasets for our method. The P, R, F1, ADD and NRD represent the precision (as %), recall (as %), F1-score (as %), average detection delay, and normalized response delay (as %) respectively. P, R, and F1 are computed using point adjustment strategy. The top-performing methods in each category have been highlighted in bold.

| Dataset | MSL | | | | | SMAP | | | | |
|---|---|---|---|---|---|---|---|---|---|---|
| Layer Number $K$ | P | R | F1 | ADD | NRD | P | R | F1 | ADD | NRD |
| 2 | 89.47 | 81.84 | 85.49 | 91.61 | 62.33 | 93.84 | 98.82 | 96.27 | 44.40 | 37.88 |
| 3 | 92.14 | **97.21** | 94.60 | 42.83 | 45.50 | **96.63** | 97.24 | **96.94** | 44.96 | 33.56 |
| 4 | **92.91** | 97.08 | **94.95** | **35.11** | **37.91** | 93.70 | **99.09** | 96.32 | **44.25** | **30.64** |
| Dataset | SWaT | | | | | PSM | | | | |
| Layer Number $K$ | P | R | F1 | ADD | NRD | P | R | F1 | ADD | NRD |
| 2 | 92.37 | **100** | 96.04 | 62.00 | 16.40 | 97.58 | 98.25 | 97.91 | 24.42 | **63.10** |
| 3 | **92.84** | 99.78 | **96.19** | 49.57 | **10.62** | **98.06** | **98.93** | **98.50** | 18.15 | 64.40 |
| 4 | 92.64 | **100** | 96.18 | **47.26** | 10.90 | 97.46 | 98.62 | 98.04 | 20.88 | 66.08 |

## I    MORE IMPLEMENTATION DETAILS

### I.1    ANOMALY RATIO SELECTION

We choose the ratio of anomaly using the Gap Statistic method, as proposed in (Tibshirani et al., 2001). Since the method is unsupervised, there are no labels in the training data. Therefore, after training the network, we compute the anomaly score for all training data. Then, we partition the anomaly scores into two distinct clusters based on their distribution. The dividing value between the two clusters is selected as the anomaly ratio for that particular dataset. For all the datasets (SWaT, PSM, SMAP, MSL), we set it to 1%. Table 18 and 19 provide details of the method.

Table 18: Anomaly ratio selection method for SWaT, MSL, and PSM dataset(the anomaly scores are normalized to 100 for ease of representation here)

| Anomaly Score | SWaT | MSL | PSM |
|---|---|---|---|
| $[0, \infty)$ | 495000 | 58317 | 132481 |
| $[0, 0.01]$ | 489528 | 57645 | 131057 |
| $(0.01, \infty)$ | 5472 | 672 | 1424 |
| Percentage in $(0.01, \infty)$ | 1.10% | 1.15% | 1.07% |

Table 19: Anomaly ratio selection method for SMAP Dataset (the anomaly scores are normalized to 100 for ease of representation here)

| Anomaly Score | SMAP |
|---|---|
| $[0, \infty)$ | 135183 |
| $[0, 0.1]$ | 133759 |
| $(0.1, \infty)$ | 1424 |
| Percentage in $(0.1, \infty)$ | 1.05% |

## J    REPRODUCIBILITY AND OPEN RESOURCES

For reproducibility, our anonymized source code is publicly available on GitHub: `https://anonymous.4open.science/r/U2AD-63A7`. The repository includes scripts and instructions to replicate our experimental findings. All datasets are publicly available or can be obtained from the original authors upon request.

## K    IMPACT STATEMENT

This paper introduces a principled framework for unsupervised anomaly detection that unifies three key components: 1) non-parametric density estimation to model nominal data patterns, 2) contextual constraints on the latent distribution using both local and global information, and 3) a controlled sampling procedure guided by a minimum-energy principle to create a clear discriminatory boundary between normal and anomalous data. The primary positive impact lies in its applicability to safety-critical domains such as network security and medical diagnostics, where its robustness to novel threats and significantly lower detection latency can prevent systemic failures.

Furthermore, as our framework is built upon diffusion models, it inherits risks common to generative models. Specifically, the model could potentially memorize sensitive or private information from its training set, which may be exposed through adversarial queries. Its generative nature could also be exploited to fabricate misleading or false information, for instance, by generating synthetic data that appears normal but contains a malicious payload.

