# OpenReview forum: "Learning Unified Representations of Normalcy for Time Series Anomaly Detection"
_ICLR.cc/2026/Conference — Submitted to ICLR 2026_

### Official Review · Reviewer_knVv · 2025-10-17

**Soundness:** 2
**Presentation:** 2
**Contribution:** 1
**Rating:** 2
**Confidence:** 3

**Summary:**

This paper proposes a diffusion-based anomaly detection framework named **U²AD**. From a high-level perspective, the framework consists of three main components. First, an SDE-based forward process adds noise to the input time series. Second, a score (diffusion) model built primarily upon attention mechanisms learns to estimate the score function. Finally, the estimated scores are used to solve an ODE in the reverse direction, reconstructing the denoised time series.
To train the framework, the authors design a set of losses, including score matching loss, reconstruction loss, two context losses, and a distributional constraint loss. Extensive experiments conducted on multiple benchmark datasets demonstrate the superiority of the proposed framework over various state-of-the-art baselines.

**Strengths:**

1. The experimental evaluation is comprehensive, covering multiple datasets and baselines, and the proposed method demonstrates strong performance.
2. The authors provide their codebase, which greatly facilitates reproducibility and further research.

**Weaknesses:**

1. **Lack of motivational grounding.**
   A large portion of the *Introduction* is devoted to describing the limitations of previous works, which I think could be condensed or moved to the appendix. More importantly, this part overlooks a significant amount of prior research on diffusion-based anomaly detection. As a result, the introduction of the proposed score-based model component does not feel well-justified. Moreover, the motivation for the contextual constraints (lines 78–80: *“a principled solution must also leverage the characteristics of each data point with respect to its adjacent neighbors as well as the global data pattern”*) is not clearly explained—how exactly does this principle apply to the anomaly detection scenario? I recommend providing illustrative examples or figures to clarify this part.

2. **Limited novelty.**
   As mentioned earlier, there already exist many diffusion-based frameworks for time-series anomaly detection [1-3]. Although the authors repeatedly emphasize that their model “learns the score function,” this is not essentially different from existing diffusion-based approaches. Apart from the SDE–ODE framework, the remaining contributions mainly lie in model design and loss formulation. In my opinion, the current presentation still feels like a combination of standard attention mechanisms and previously known components. I suggest placing more emphasis on the conceptual design of the model and losses, and explaining their underlying rationale more thoroughly.

3. **Methodological flaws.**
   These issues mainly concern the loss design. After a brief inspection of the released code, I have several concerns:

   * The **Reconstruction Loss** always requires backward sampling every time (independent of $t$ in the code implementation), which I find rather *inelegant* and implies potential inefficiency in training.
   * The **Volume Minimization Objective** may overly constrain the diffusion model during training. This seems contradictory to the paper’s stated goal of encouraging the model to generalize well and detect unseen anomalies.
   * The **Contextual Information Gain** term is unclear—why is it appropriate to impose a KL-divergence constraint between local and global features of different types? Even inter-layer feature interactions would be a more convincing formulation.

4. **Lack of experimental rigor.**
   From the reported F1 scores, it is evident that the authors used *point-adjusted* evaluation (F1-PA). However, even random guessing can achieve high F1-PA values depending on the data distribution [4], so this metric alone is insufficient. In addition, as I suspected, there is no analysis of computational complexity (training, inference time, or parameter count), which is essential for fair comparison.

**Reference:**

[1] Xiao, Chunjing, et al. "Imputation-based time-series anomaly detection with conditional weight-incremental diffusion models." Proceedings of the 29th ACM SIGKDD conference on knowledge discovery and data mining. 2023.
[2] Chen, Yuhang, et al. "ImDiffusion: Imputed Diffusion Models for Multivariate Time Series Anomaly Detection." Proc. VLDB Endow. (2023).
[3] Hu, Rongyao, et al. "Unsupervised anomaly detection for multivariate time series using diffusion model." ICASSP 2024-2024 IEEE International Conference on Acoustics, Speech and Signal Processing (ICASSP). IEEE, 2024.
[4] Kim, Siwon, et al. "Towards a rigorous evaluation of time-series anomaly detection." Proceedings of the AAAI conference on artificial intelligence. Vol. 36. No. 7. 2022.

**Questions:**

See the Weaknesses section.

---

### Official Review · Reviewer_JHrb · 2025-10-17

**Soundness:** 2
**Presentation:** 3
**Contribution:** 2
**Rating:** 2
**Confidence:** 3

**Summary:**

The paper proposes U2AD, a unified unsupervised framework for time series anomaly detection based on score-based generative modeling. It learns normal data distributions through a dual-pathway score network and a unified training objective combining score matching, contextual, and reconstruction losses. Experiments on benchmark datasets show that U2AD achieves state-of-the-art accuracy and detects anomalies earlier than existing methods.


The idea of combining global and local representations was proposed years ago (see PUAD for reference), so the authors should better highlight the advantages and distinctions of their work.

More recent methods should be included in the experiments to further validate the effectiveness of the proposed approach.

The authors need to clearly specify the model’s innovations and theoretical advantages, as the dual-pathway score network alone may not be sufficient as the core contribution of an ICLR paper.

It would be helpful to include qualitative experiments demonstrating that the proposed model is indeed effective—for example, showing that the global pathway truly captures global information—rather than just presenting a collection of neural networks without evidence that they capture what is claimed in the paper.

**Strengths:**

see summary

**Weaknesses:**

see summary

**Questions:**

see summary

---

### Official Review · Reviewer_7Qcr · 2025-10-27

**Soundness:** 2
**Presentation:** 1
**Contribution:** 2
**Rating:** 2
**Confidence:** 4

**Summary:**

This paper proposes a score-based generative framework for time-series anomaly detection that aims to learn a unified representation of normal data through contextual information gain. Extensive experiments and ablation studies on multiple benchmark datasets demonstrate the model’s effectiveness compared to existing approaches.

**Strengths:**

The experimental section is comprehensive. The authors conduct a large number of comparative and ablation studies across multiple datasets, providing empirical evidence for the model’s effectiveness.

**Weaknesses:**

1. Poor writing and unclear motivation. The overall presentation is confusing. The paper does not clearly articulate the limitations of existing methods nor distinguish its own contributions. For example, the issues mentioned in the abstract and introduction (“they struggle to learn a robust representation...” and “prevailing paradigms... built on simplifying assumptions”) are too vague, and the claimed contribution (“learns the underlying data distribution of normal samples by utilizing score-based generative modeling”) does not clarify how this differs from prior score-based methods.

2. Figures 1 and 2 are unclear and uninformative. Figure 1, placed in the introduction, shows types of anomalies but has no direct connection to the proposed method and is not explained in the text. Figure 2 seems to illustrate reconstructed anomalies overlapping with normal data, which appears counterintuitive. The paper does not clarify why such reconstruction is beneficial or what insight this visualization provides.

3. Confusing definition of Contextual Information Gain. The description that Contextual Information Gain “maximizes the boundary based on local characteristics and minimizes it based on global characteristics” suggests an adversarial mechanism, but Equation (10) does not reflect this idea. It is unclear how minimizing this loss achieves the intended local/global balance or the separation between normal and anomalous samples.

4. Inconsistency in the deterministic reverse solver. The derivation from Equation (3) to Equation (8) simply removes the last term, but the paper does not justify how this modification preserves consistency with the forward diffusion (noise-adding) process.

5. Questionable rationale in Volume Minimization Objective. The center c is defined as the “center of the data distribution in latent space.” However, forcing noise gradients toward a central point of the latent space may not meaningfully improve anomaly discrimination.

**Questions:**

See the weaknesses section.

---

### Meta-Review · Area_Chair_N9aa · 2026-01-07

**Summary:**

This paper proposes $U^2AD$, a score-based generative (diffusion) framework for multivariate time-series anomaly detection that learns a representation of “normalcy” via a time-dependent score network, multiple contextual constraints, and ODE-based deterministic reconstruction. Reviewers consistently acknowledged the comprehensive experiments and that results appear strong on several benchmarks, with released code supporting reproducibility. However, all three reviews converge on major weaknesses that drive rejection: (i) unclear motivation and positioning, with vague claims about prior limitations and insufficient engagement with existing diffusion-based anomaly detection literature; (ii) limited novelty, as the method is viewed as a combination of known diffusion/SDE–ODE components with attention-based architecture and loss engineering, without a clearly articulated core insight; (iii) methodological clarity and soundness issues, including unclear definitions and rationale for “Contextual Information Gain,” questionable justification for the deterministic reverse solver modification, and concerns that the volume-minimization objective may be poorly motivated or overly constraining; and (iv) evaluation rigor gaps, notably reliance on point-adjusted F1 (F1-PA) without broader metrics/analysis, and missing computational complexity reporting (training/inference time, parameter count) needed for fair comparison.

**Reviewer Scores:**

All three reviewers are solid rejects. Given that their critiques focus on novelty/positioning and methodological soundness rather than minor presentation issues, fuller discussion alone would be unlikely to change scores meaningfully without substantial new evidence.

---

### Decision · Program_Chairs · 2026-01-26

Reject